# Towards the application of Stokes flow equations to structural restoration simulations

**Melchior Schuh-Senlis[1], Cedric Thieulot[2], Paul Cupillard[1], and Guillaume Caumon[1]**

[1]Université de Lorraine, CNRS, GeoRessources, France- 54000 Nancy, France
[2]Utrecht University, Netherlands

**Correspondence:** Melchior Schuh-Senlis (melchior.schuh-senlis@univ-lorraine.fr)

**Abstract.**

Structural restoration is commonly used to assess the deformation of geological structures and to reconstruct past basin geometries. For this, geomechanical restoration considers faults as frictionless contact surfaces. To bring more physical behavior and better handle large deformations, we build on a reverse time Stokes-based method, previously applied to restore salt structures with negative time step advection. We test the applicability of the method to structures including sediments of variable viscosity, faults and non-flat topography. We present a simulation code that uses a combination of Arbitrary Lagrangian-Eulerian methods and Particle-In-Cells methods, and is coupled with adaptive mesh refinement. It is used to apply the reverse time Stokes-based method on simple two-dimensional geological cross-sections and shows that reasonable restored geometries can be obtained.

## 1 Introduction

The Earth's subsurface is the result of millions of years of deformation. Determining the deformation history from present-day structures has been a concern for geoscientists who try to understand and quantify basin evolution. Restoration is an ensemble of methods which allow such quantification, by reversing processes that led to the current geometry of a geological region (e.g., Chamberlin, 1910; Dahlstrom, 1969). It covers a number of different processes and methodologies. The classical techniques are unfolding and unfaulting using length/area preservation in order to remove the effects of tectonic forces. In addition to this, several methods have been developed to take into account the effects of other important parameters, like erosion and deposition of sediments (e.g., Dimakis et al., 1998), isostasy

compensation (e.g., Allen and Allen, 2013), thermal subsidence due to mantle thermal effect (Royden and Keen, 1980; Allen and Allen, 2013), rock decompaction due to a change of load (e.g., Athy, 1930; Durand-Riard et al., 2011; Allen and Allen, 2013), or, at a smaller scale, the reverse migration of channelized systems (e.g., Parquer et al., 2017). These methods allow us to evaluate the consistency of a model and test the hypotheses which lead to its construction, in order to generate paleo-basin geometries consistent with present-day observations for use in more elaborate hydro-mechanical forward models (e.g., Bouziat et al., 2019). In this article, we focus on the structural restoration based on unfolding and unfaulting.

Since the beginning of the last century, unfolding and unfaulting has been mostly done with geometric and kinematic rules (e.g., Chamberlin, 1910; Dahlstrom, 1969; Gratier, 1988; Rouby, 1994; Groshong, 2006; Lovely et al., 2018; Fossen, 2016). The first implementations in two dimensions (2D) used balanced restoration, which relies on the conservation of layer bed area and thickness (e.g., Chamberlin, 1910; Dahlstrom, 1969; Groshong, 2006). Map restoration was then developed to study deformations which are mainly horizontal; it can be qualified as a 2.5D method (e.g., Cobbold and Percevault, 1983; Rouby, 1994; Ramón et al., 2016). Later, three dimensional (3D) geometrical methods have been proposed (Massot, 2002; Muron, 2005; Lovely et al., 2018), allowing the tracking of internal volumetric deformation. Such methods are all based on the minimization of horizon deformation and on volume conservation, and therefore considerably simplify rock deformation mechanisms, ignore mechanical layering effects and are limited when considering salt basins. In this light, numerous authors have stressed out the necessity of

incorporating more physical principles into the restoration of geological models (Fletcher and Pollard, 1999; Ismail-Zadeh et al., 2001; Muron, 2005; Maerten and Maerten, 2006; Moretti, 2008; Guzofski et al., 2009; Al-Fahmi et al., 2016).

Several solutions have emerged for volumetric mechanics-based restoration, that differ in terms of computational techniques and scale of the area of interest. These solutions can be divided in two main approaches, that have been developped to adress two different problematics of restoration. They differ both in the mechanical laws used to compute the motion of rock layers, and in how these mechanical laws are applied to restore geological models.

The first approach considers the restoration of sediment layers assumed to deform elastically between frictionless fault surfaces. It has been developed since the 2000s as a geomechanical simulation with specific boundary values (Maerten and Maerten, 2001; De Santi et al., 2002; Muron, 2005; Moretti et al., 2006; Maerten and Maerten, 2006; Guzofski et al., 2009; Durand-Riard et al., 2010, 2013a, b; Tang et al., 2016; Chauvin et al., 2018). In this approach, internal deformation is not known *a priori*, and the strain is computed from the mechanical behavior of rocks and the applied boundary conditions. The model is parameterized with elastic properties to mimic the response of rocks to mechanical stresses and the restoration displacement is computed by solving the equation of motion, in which the Cauchy stress tensor is defined by Hooke's law. The restoration itself is performed by applying specific boundary conditions to constrain the model. These conditions, usually imposed on the displacement, rely on the following assumptions: the uppermost horizon was flat and horizontal at deposition time, and it was not faulted. Other conditions can be introduced as complementary geological knowledge, such as direction and scale of deformation, or amount of lateral displacement (Chauvin et al., 2018). Although these methods offer significant advances in the structural restoration of geological models, they still present many limitations. First, the boundary conditions set to unfold and unfault the medium are unphysical as the imposed depth of the free surface is the main driver of the deformation (Lovely et al., 2012; Chauvin et al., 2018). These conditions are convenient hypotheses which do not necessarily reflect the paleo-stress state, hence they can be questionned (Durand-Riard et al., 2010; Lovely et al., 2012; Durand-Riard et al., 2013a). Secondly, geomechanical restoration so far only considers elastic rock properties, neglecting other possible behaviors, such as viscous, visco-elastic or plastic deformation (Gerbault et al., 1998). Transverse isotropic behavior also affects strain localization during restoration (Durand-Riard et al., 2013a), but such a behavior is rarely applied in practice. These physical issues raise the question of the capability of this restoration approach to properly recover paleo-deformation. As a consequence, there are no clear guidelines on which method to choose between geometric and kinematic restoration and geomechani-

cal restoration, despite the more physical approach of the second one (Maerten and Maerten, 2006; Guzofski et al., 2009). Moreover, in spite of its name, geomechanical restoration is extensively controlled by geometric considerations: flattening of the top layer and a geometric unfaulting based on frictionless contact conditions to stitch the horizon cutoff lines accross each fault. Another practical issue is the need for a valid volumetric mesh of the structural model, including a boundary representation of the geological domain with the horizons and faults as boundaries (e.g., Muron, 2005), even if the use of implicit horizons relaxes this constraint (Durand-Riard et al., 2010). Such a mesh is difficult to generate, as shown for example by Pellerin et al. (2014), Zehner et al. (2016) and Anquez et al. (2019). Since restoration deals with large deformations, the model evolves and may need to be remeshed. The remeshing algorithms, however, are limited because key structural elements like faults and horizons must be preserved for geomechanical restoration to be used as an interpretation validation tool. To sum up, this restoration approach has overcome some limitations of the "classical" geometric restoration process, by taking some of the internal movement of the layers into account for example, but it still needs to be improved to better account for different rheologies, larger deformations, faults, salt tectonics, and boundary conditions.

The second approach was introduced in 1999 as a way to improve the restoration of salt structures (Kaus and Podladchikov, 2001; Ismail-Zadeh et al., 2001, 2004; Ismail-Zadeh and Tackley, 2010). It relies on considering the rocks as viscous fluids to compute the motion, and applying negative timesteps. It is motivated both by the fact that rock salt and some sediment overburdens behave as viscous fluids over time-scales of millions of years, and by the reversibility of the Stokes equations, which allow the backward timestepping. The first implementations used a linear viscous (Newtonian) rheology, and proved to be able to restore 2D seismic cross-sections of salt diapirs (Ismail-Zadeh et al., 2001), and 3D Rayleigh-Taylor instabilities (Kaus and Podladchikov, 2001; Ismail-Zadeh et al., 2004). Since then, the method has been used for 3D unfolding in the absence of gravity (e.g., Schmalholz, 2008), extended to non-linear (power-law) viscous behavior (e.g., Lechmann et al., 2010; Fernandez Terrones), or used to study the reverse modelling of flanking structures (e.g., Kocher and Mancktelow, 2005). Overall, this approach has proven to allow the unfolding of sediment layers and the restoration of salt structures, both in 2D and in 3D. In the various previous applications, however, faults are either not present or not taken into account in the restoration process. Also, the top surface in contact with air stays flat during the restoration process as the sedimentation and erosion process are mostly considered fast enough to flatten the arising topography.

In this paper, we investigate a way of addressing some of the challenges raised by the first approach. We show that it is possible to push the second approach further and apply it

to models with faults and a non-flat free surface. For simplicity, we neglect the influence of temperature and consider the rocks as having a (linear) Newtonian rheology. While this considerably simplifies any non-Newtonian or visco-elasto-plastic behavior in rocks, we show in the manuscript that this consideration is sufficient in simple setups. In the case of more complex overburdens, the method proved in the litterature to be able to restore various structures with power-law viscosities (e.g., Lechmann et al., 2010; Fernandez Terrones). We introduce a numerical scheme combining features of the Arbitrary Lagrangian-Eulerian (ALE) and Particle-In-Cell (PIC) approaches, and using adaptive grid refinement. This specific implementation is motivated by the need for a moving topography, as well as the high accuracy needed for the computation of motion around the faults (). We show that this scheme is accurate enough to consistently restore various geological setups, including faults.

The outline of this paper is as follows: we first present the concepts of Stokes flow-based restoration and its physical underpinnings. In a second part, we introduce the numerical code we developed for this application. Finally, we show the results that were obtained on an upscaled version of the model presented by van Keken et al. (1997), on a model with no prior knowledge on the material properties and boundary conditions to apply, and on a model with faults and a non-flat free top surface.

## 2 Using creeping flow equations for geomechanical restoration

### 2.1 Creeping flow equations

The standard equations for creeping flows are the Stokes equations, consisting of the momentum conservation equation

$$\boldsymbol{\nabla} \cdot \boldsymbol{\sigma} + \boldsymbol{f} = \boldsymbol{0} \tag{1}$$

and the mass conservation equation for incompressible fluids (continuity equation)

$$\boldsymbol{\nabla} \cdot \boldsymbol{v} = 0, \tag{2}$$

where $\boldsymbol{\nabla}$ is the del operator, $\boldsymbol{\sigma}$ is the stress tensor, $\boldsymbol{f}$ is the specific body force (usually the volumetric weight $\rho \boldsymbol{g}$), and $\boldsymbol{v}$ is the velocity. The stress consists of a deviatoric part $\boldsymbol{\tau}$ and an isotropic pressure $p$:

$$\boldsymbol{\sigma} = \boldsymbol{\tau} - p\mathbf{I}, \tag{3}$$

where $\mathbf{I}$ is the identity tensor. In the viscous flow assumption, the deviatoric part of the stress is

$$\boldsymbol{\tau} = 2\eta\mathbf{D}, \tag{4}$$

with $\eta$ the dynamic viscosity and $\mathbf{D}$ the infinitesimal strain rate tensor defined by

$$\mathbf{D} = \frac{1}{2}\left[\boldsymbol{\nabla}\boldsymbol{v} + (\boldsymbol{\nabla}\boldsymbol{v})^T\right]. \tag{5}$$

Assembling Eq. (1), (3), (4) and (5), the momentum conservation equation can be written

$$\boldsymbol{\nabla} \cdot \left[\eta(\boldsymbol{\nabla}\boldsymbol{v} + (\boldsymbol{\nabla}\boldsymbol{v})^T)\right] - \boldsymbol{\nabla}p = -\rho\boldsymbol{g}. \tag{6}$$

Here, we deal with materials that are highly viscous (with a viscosity $\eta$ over $10^{17}$ Pa.s), over time scales of thousands to millions of years, so these equations neglect the inertial part of the Navier-Stokes equations (Massimi et al., 2006). As such, they describe a steady-state flow and their resolution provides the velocity of a fluid at a specific position and time. When different fluids are present, the conditions that are applied at their boundaries, as well as their differences in density, can create instabilities such as Rayleigh-Taylor instabilities. These instabilities make the flow non-stationary as they advect the viscosity and density fields in time.

### 2.2 Restoration idea

In forward simulation schemes, the Stokes equations (6) and (2) are solved for pressure and velocity, and the material representation of the geological model is advected from the velocity at each time step. The simplest way to do it is by using an Euler scheme, the position $\boldsymbol{x}(t + \Delta t)$ of each point of the material model after one time step being computed as

$$\boldsymbol{x}(t + \Delta t) = \boldsymbol{x}(t) + \boldsymbol{v}(t) \cdot \Delta t, \tag{7}$$

with $\boldsymbol{x}(t)$ the position and $\boldsymbol{v}(t)$ the computed velocity of the point at time $t$, and $\Delta t$ the time step (while higher-order methods exist (e.g., Ismail-Zadeh and Tackley, 2010), particularly to stabilize the advection scheme in the case of large time steps, we choose to present the restoration idea with this one for simplicity). This Finite-Difference approximation relies on the idea that if the chosen time step $\Delta t$ is small enough, we can approximate the velocity of a particle as a constant over this time step ($\Delta t$ is usually calculated using a Courant-Friedrichs-Lewy (CFL) condition (Courant et al., 1928) to ensure it). Since the Stokes equations are linear and do not depend on previous time steps for the computation of the velocity, we can extend this approximation to backwards simulations. This is the basis of backward time stepping restoration schemes: instead of applying Eq.(7), we apply

$$\boldsymbol{x}(t - \Delta t) = \boldsymbol{x}(t) - \boldsymbol{v}(t) \cdot \Delta t \tag{8}$$

for the advection of the points of the material model, at each time step, like in Fig. 1.

In this light, using viscous fluid properties instead of elastic properties to represent the mechanical behavior of geological materials holds several advantages, such as the use of boundary conditions that are closer to reality, like a free surface on top, or the account of other rheologies like a salt layer.

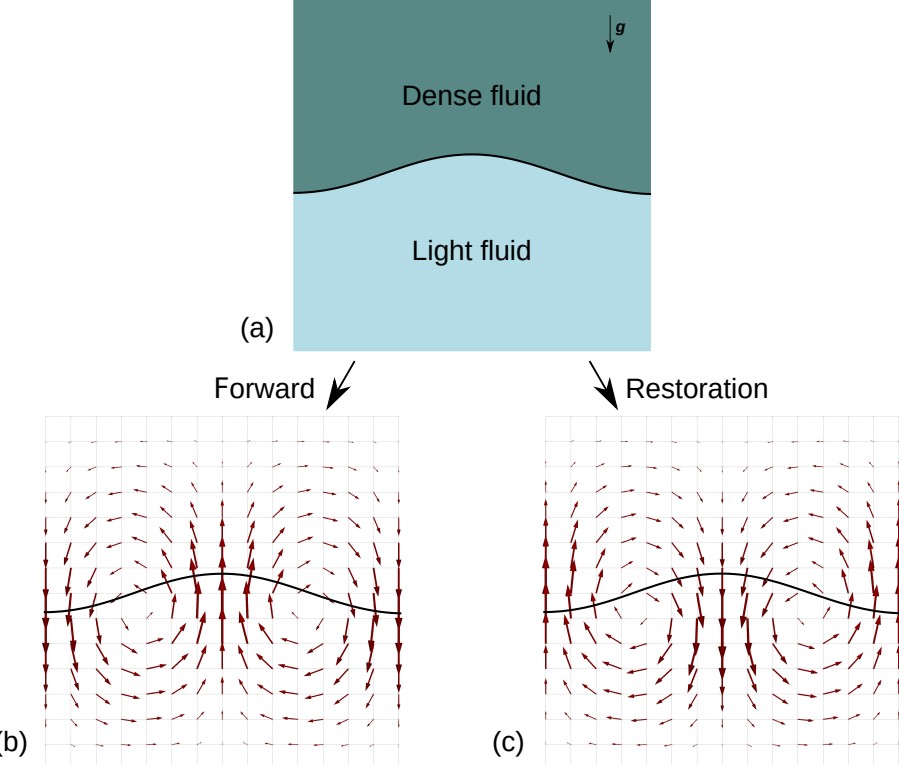

**Figure 1.** Example of the restoration scheme for a simple setup (a): as the arrows in (b) represent the velocity computed at a specific time step for a forward scheme, the advection of the material model in a restoration scheme is done with the opposite of the computed velocity, shown in (c).

## 3   Implementation in a specific code

### 3.1   Presentation

The restoration scheme presented in Sect. 2 has been implemented in the FAIStokes[1] code. It relies almost entirely on the deal.II library (Bangerth et al., 2007; Arndt et al., 2019, 2020) for all Finite Element related algorithms. The material tracking is based on the Particle-In-Cell (PIC) method (e.g., Asgari and Moresi, 2012; Thielmann et al., 2014; Gassmöller et al., 2016, 2018, 2019; Trim et al., 2019). The general work-flow of the code is shown in Fig. 2 and details of implementation are discussed in the following sub-sections. Five benchmarks have been carried out to test the computation parts of the code and are presented in Appendices A, B, C, D and E.

### 3.2   Finite Element discretization

The Finite Element Method (FEM) was introduced in the late 1950's (Hughes, 2012). Since then, it has emerged as one of the most powerful methods for solving Partial Differential Equations (PDEs) numerically. In FAIStokes, the FEM algorithms are based on the deal.II library. The do-

main is discretized on a set of quadrilateral elements, on which Finite Element (FE) basis functions are defined. The aim of this paper is not to do a thorough review of the FEM, so only the specifications of the FAIStokes code will be presented here. For solving the Stokes equations, we use quadrilateral Taylor-Hood $Q_2 \times Q_1$ elements that satisfy the Ladyzhenskaya-Babuška-Brezzi (LBB) condition for stability (Donea et al., 2004). Contrarily to many creeping flow codes that are used to study the subsurface, we do not solve the heat transport equation, both for simplicity and because it is likely to have only a small effect on the strain at the scale at which structural restoration is generally applied (i.e. basin-scale, close to the surface). Moreover, there may be important temperature diffusion at geological time scales, particularly in salt layers, and it is not reversible. We use Dirichlet and Neumann boundary conditions that we adapt (e.g. rigidity, free-slip, free surface, specific traction or velocity) for each boundary to the different problems at hand. Appendices A, B, D and E showcase results of the FE benchmarking.

### 3.3   Material discretization

The geomechanical simulation of a specific domain requires to choose an appropriate kinematic description to follow the displacement inside the geological layers. Continuum me-

---

[1]**F**inite element **A**rbitrary Eulerian-Lagrangian **I**mplementation of **Stokes**

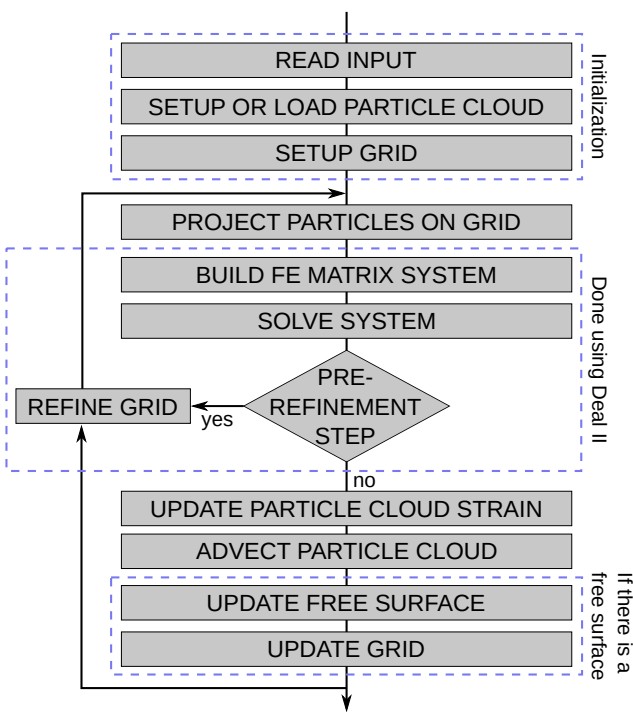

**Figure 2.** Schematic workflow of the FAIStokes code structure. The pre-refinement step occurs at the beginning of the simulation (or during a reinitialization of the grid) to ensure that the velocity used for the advection step is computed using the adaptively refined grid.

chanics first distinguished two main frames: the Eulerian frame of reference, also known as the spatial description, and the Lagrangian frame of reference, also known as the material description (Cornet, 2015). Both methods have their
5 advantages and disadvantages, but neither of them is specifically adapted in the case of large displacements over time, such as those studied here. In order to overcome the limitations of the two approaches, the Arbitrary Lagrangian Eulerian (ALE) formulation (Fullsack, 1995; Donea et al., 2004),
10 which inherits features from both methodologies, was developed. It has various formulations and implementations, both in 2D (e.g., Willett et al., 1993; Poliakov et al., 1996; Massimi et al., 2006, 2007; Fillon et al., 2013; Rose et al., 2017) and, more recently, in 3D (e.g., Braun, 2003; Thieulot, 2011;
15 Thieulot et al., 2014). Most of these methods rely on keeping track of the material properties in a Lagrangian way, while computing the displacement on a grid that can only deform vertically to account for a possible free surface. It is particularly useful in geomechanics, where the vertical deformation
20 is generally small compared to the horizontal deformation, and in the case of highly viscous fluids in the mantle, for which the density and viscosity depend mostly on the temperature and depth. In FAIStokes, the grid has an ALE part as it can adapt to follow the movement of the free surface.

### 3.4 The PIC method

During mechanical simulations, the material properties inside the model are tracked using particles; each of these particles discretize the small part of the model around them and its properties. At each time step, the material properties of the particles are projected onto the grid. They are then used to solve the Stokes equations on the grid. Following this, the particles are advected using the solution on the grid.

At the begining of the simulations, FAIStokes either creates a model from a function giving the distribution of the material parameters or loads a particle swarm from a file. In the first case, a regularly distributed particle swarm is generated, with a density of particles depending on the size of the smallest element of the computation grid. The given function is then used to associate the material properties to the particles depending on their position. Since the particle swarm doesn't directly track the interfaces, it has to be dense enough to recover accurately the material properties of the model; depending on the simulation, some parts of the model can therefore be densified to keep the appropriate accuracy. At each time step, the material properties are interpolated from the particle swarm to the grid in order to build the FE matrix and its preconditioner. For each element, the density is interpolated on the quadrature points using an arithmetic mean of the densities of the particles around the quadrature points (closer than a distance depending on the smallest element of the domain). The viscosity is recovered for each element

using a harmonic mean of the viscosities of the particles inside the element. This reduces the effect of very high viscosity differences (possibly of several orders of magnitude) on the solver and is more computationally efficient despite the higher grid refinement needed (Deubelbeiss and Kaus, 2008; Thielmann et al., 2014; Heister et al., 2017). In the simulations we present hereafter, we were able to verify that this averaging verifies the conservation of the volume and mass in the model. Appendices A, B, D and E test the interpolation of the material properties from the particle swarm to the finite element grid to reasonable accuracy.

## 3.5  Grid and solvers

The grid and solvers come from the deal.II code, and their use is highly inspired from the deal.II tutorials $step-31$ [2] and $step-32$ [3]. The grid is created first as a quadrilateral from the coordinates of the bottom left and top right corners of the domain. This quadrilateral is then split in order to get cells closest to a square (depending on the model bounding box size) and refined and coarsened adaptively several times to construct the initial grid. The FE matrix, its preconditioner and the right-hand side force-vector are constructed using the material properties interpolated from the particle swarm as described in the previous subsection. In the right-hand side, the norm of the gravity vector $g$ of Eq. (6) is always $9.81$ m.s$^{-2}$ in our simulations, and its direction is always downwards. The matrix system is solved using an iterative FGMRES solver preconditioned by a block matrix involving the Schur complement (Kronbichler et al., 2012). This solution is then used to refine and coarsen the grid adaptively using deal.II's features, based on a gradient estimator in order to minimize the local error. Depending on the input level of refinement, the cycle of building the matrix system, solving it, and adaptively refining and coarsening the grid is repeated several times, as shown in Fig. 2. Appendices A, B, D, E show the results of benchmarks that tested the computation of the velocity on different setups.

## 3.6  Velocity interpolation

Once the grid refinement has been completed, the particle swarm is advected by the obtained solution. In FAIStokes, the interpolation of the velocity is done separately in each grid cell with a $Q_2$ interpolation scheme. Depending on whether the simulation is forward or backward, the displacement of each particle for a time step $\Delta t$ is computed using Eq. (7) or (8). The value of $\Delta t$ is computed from the CFL condition. The default value for the CFL number is $0.085$, but it can be reduced depending on the simulation (for example, the results shown in the next section use a CFL number of $0.0085$, while the benchmarks in the Appendix use a CFL number of $0.042$). The advection is done with a $2^{nd}$-order

___________

[2]https://dealii.org/9.0.0/doxygen/deal.II/step_31.html
[3]https://dealii.org/9.0.0/doxygen/deal.II/step_32.html

Runge-Kutta scheme in space: at each time step, the particles are first advected by half the computed displacement; the velocity is then interpolated on their new position to update the displacement, and particles are advected them again by half of this new displacement. This scheme reduces the error in the advection process without need for simulation time step refinements. It is computationally efficient because the computation of the displacement on the particle swarm is inexpensive as compared to solving the FE matrix system. Appendices C, D, E show the results of benchmarks that tested the interpolation of the velocity in time-dependant problems.

## 3.7  Free surface implementation

In the case of a free surface on the top of the model, the top surface is tracked by a separate point swarm. This point swarm is denser than the material particle swarm and is one dimension lower (i.e. a line in our 2D cases). It is advected at each time step the same way as the particle swarm that represents the geological model. After its displacement or during the setup of the grid, the free surface point swarm is used as a reference to move vertically the nodes of the grid at the top of the model, so that they match the free surface. This vertical displacement is then propagated to the rest of the grid so that the grid cells stay as close to squares as possible, while not affecting the other boundaries. Fig. 3 illustrates the whole process. Since our models are isothermal, no special processing is required to correct the temperature field during this process. Appendix D shows the results of a benchmark that tests the free surface implementation along with other computational parts of the code. The free surface stabilization algorithm (refered to as FSSA in the rest of the paper) developed by Kaus et al. (2010) and showcased in Quinquis et al. (2011) has been implemented in FAIStokes; we benchmark it in Appendix E.

## 4  Results

In addition to the benchmarks presented in the Appendices, which mainly check the algorithms of the code, we tested our specific restoration scheme on three simple models. In those experiments, the boundary conditions are simplified and quite unrealistic, but the goal here is to check the behavior of the reverse-time modeling in simple settings. In particular, we choose to neglect basal and lateral displacements in the first two models, which are known to play a role in salt tectonics (Koyi, 1996; Ismail-Zadeh et al., 2004) but would require a calibration and would increase the degrees of freedom of the problem.

## 4.1  Diapiric growth model

The first model is scaled-up from van Keken et al. (1997). The setup consists of a simple two-layered system driven by gravity, as shown in Fig. 4. The upper layer represents

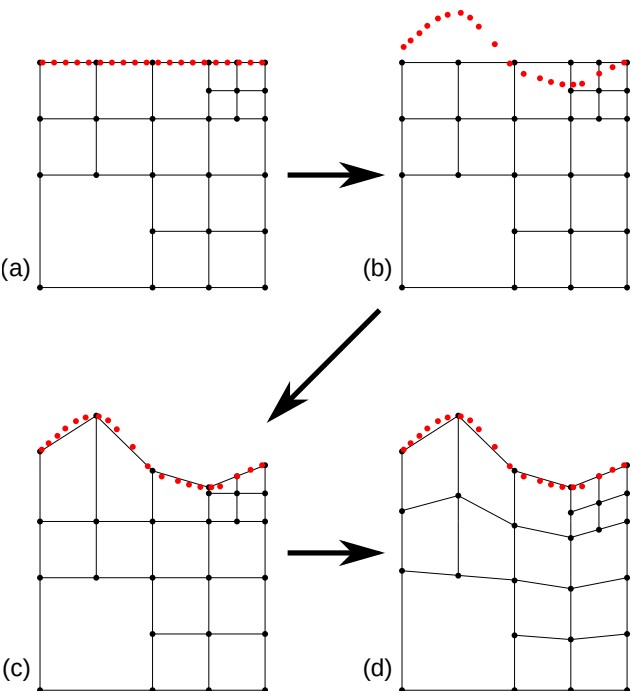

**Figure 3.** Process for the update of the free surface (the motion is exagerated for the sake of the explanation, and is less extreme in reality): (a) Initial state where the velocity is computed on the grid. (b) The point swarm tracking the free surface is advected according to the computed velocity. (c) The grid nodes at the top of the free surface are moved vertically to match the point swarm. (d) The deformation of the grid is diffused to the rest of the nodes.

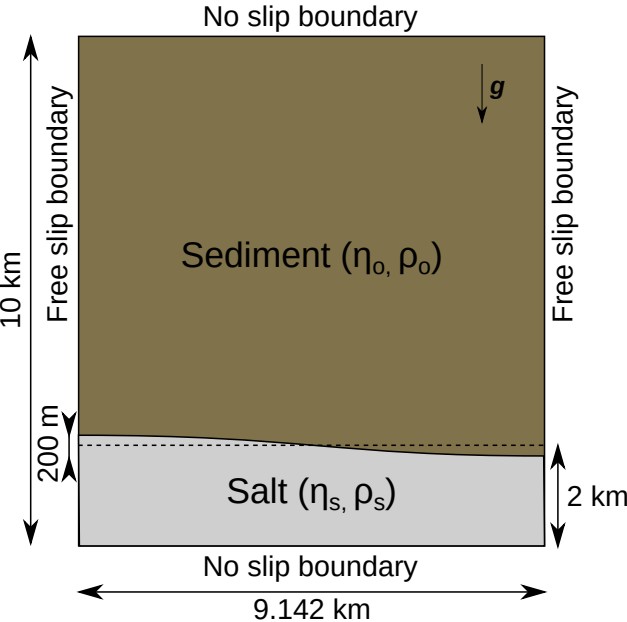

**Figure 4.** Setup of the model scaled-up from van Keken et al. (1997).

sediments that are denser than the lower layer which contains salt ($\rho_o = 2600$ kg.m$^{-3}$ for the sediment layer and $\rho_s = 2150$ kg.m$^{-3}$ for the salt layer). A sinusoidal instability initiates the movement at the begining of the simulation.

The model is limited to a $10$ km $\times 9.142$ km domain (the width value is given by van Keken et al. (1997) to yield the largest growth rate for the diapir) with free slip boundary conditions on the sides and no slip boundary conditions on

the top and bottom sides. The grid has $32^2$ initial elements and two levels of additional adaptive refinement. The particle swarm has a heterogeneous particle density: it is first sampled regularly in the model and then densified to five times more particles around the interface between the two layers to facilitate the tracking of material properties. The average distance between two particles near the interface is $14.3$ m. The total number of particles is $64,000$. Two experiments were performed in this model: the first one as a test with isoviscous materials ($\eta_o = \eta_s = 10^{19}$ Pa.s), the second one with material properties closer to reality with a lower viscosity for salt ($\eta_o = 2.8 \times 10^{19}$ Pa.s for the sediment layer and $\eta_s = 1.4 \times 10^{17}$ Pa.s for the salt layer).

For each experiment, we first did a forward simulation, and then we applied the restoration scheme to the results obtained at the end of the simulation. The state obtained after $6 \times 10^6$ years for the first test and $1.5 \times 10^6$ years for the second test, as well as the restored models, are shown in Fig. 5. We can see that while the isoviscous experiment has a rather smooth forward result, the second experiment with a less viscous salt leads to the creation of a salt weld (surface where the salt layer thickness has reached or almost reached zero, the salt having creeped away) at the bottom and left-hand side of the model.

To check the quality of the restoration in the two experiments, we compute for each particle the distance between its original position before the forward simulation and its position at the end of the restoration process. The mean value for this distance is $14$ m ($0.1\%$ error) for the isoviscous case and $201$ m ($2\%$ error) for the variable viscosity case, and the maximum value is $143$ m ($1.5\%$ error) for the isoviscous case and $4947$ m ($49\%$ error) for the variable viscosity case. While these results are quite good for the isoviscous case, we could think that the variable viscosity case restoration is too inaccurate. Histograms for the errors in the two experiments are given in Fig. 6 and Fig. 7, and help explain this phenomenon. The high error values in the variable viscosity case are due to the creation of a basal weld, which mixes the particles at the bottom of the model. Some of these particles are not well restored and stay at the bottom of the model, creating very large errors (hence the error bars of 1 to 20 particles with an error higher than $500$ m in Fig. 7). The basal weld in itself creates large distortions which explain the overall large errors at the interface. However, if we look at the model at the end of the experiments in a global way, not taking into account small irregularities, and study only the boundary between the two layers, the maximum distance between the initial model and the restored model is only $50$ m ($0.5\%$ error) for the first experiment and $125$ m ($1.25\%$ error) for the second, which is acceptable considering the large amount of total deformation.

## 4.2 Stochastically generated salt diapir model

This model was generated with the method proposed by Clausolles et al. (2019). It consists of a salt diapir that mimicks passive diapirism structures created by syndeformation differential sediment loading. The input for the salt diapir is a seismic image interpreted to segment it in three regions: salt, sediment, and uncertain. The salt-sediment interface is then generated in the uncertain zone, from available data, geological knowledge and a random scalar field that takes into account the uncertainties. The setup is quite simple but interesting for two reasons. First, this model was not created by a forward viscous simulation, and the rheology of the salt and sediments is not known. Second, this model has a high uncertainty and it is uncertain wether the boundary conditions we apply can restore it or not. Therefore, this test case can be assimilated to the simplification of a real case application. The initial particle swarm contains $102,510$ particles regularly sampling the model, and we apply free-slip boundary conditions on the top and side model boundaries, and a no slip boundary condition on the bottom. Figure 8 shows the initial state of the model. The grid has $48 \times 80$ initial elements and three levels of additional adaptive refinement; its state at the beginning of the simulation is shown in Fig. 9. In order to assess the influence of the value of the parameters on the results of the restoration, we tested different possibilities. For the density, the value for salt rock is $\rho_{salt} = 2160$ kg.m$^{-3}$, while the value for sediments can vary depending on the type and origin of deposition mechanisms; we considered here a value $\rho_o \in [2600; 3300]$ kg.m$^{-3}$. For simplicity, we set the viscosity of the salt layer at $\eta_{salt} = 10^{17}$ Pa.s and only vary the viscosity of the sediments $\eta_o \in [10^{19}; 10^{21}]$ Pa.s in order to test the effect of the constrast.

We did five experiments with different values of $\rho_o$ and $\eta_o$:

– Exp.1: $\rho_o = 2600$ kg.m$^{-3}$, $\eta_o = 10^{19}$ Pa.s

– Exp.2: $\rho_o = 3300$ kg.m$^{-3}$, $\eta_o = 10^{19}$ Pa.s

– Exp.3: $\rho_o = 2950$ kg.m$^{-3}$, $\eta_o = 10^{20}$ Pa.s

– Exp.4: $\rho_o = 2600$ kg.m$^{-3}$, $\eta_o = 10^{21}$ Pa.s

– Exp.5: $\rho_o = 3300$ kg.m$^{-3}$, $\eta_o = 10^{21}$ Pa.s

As this is a simplification of a real case application, and there is no information on the type of sediments, in each experiment the density and viscosity are homogeneous in the sediment and salt layers.

The results for the 5 experiment simulations are given in Fig 10. Depending on the experiment, we choose to stop the restoration process after different durations $t_{end}$. Indeed, as the viscosity and density vary from one experiment to the other, so does the model relaxation time.

Overall, the restoration process removes the diapir and leaves a salt scar, while the sediment layers remain globally flat. Since this setup is generated by a method for syndeformation diapirs, a full restoration of the model should

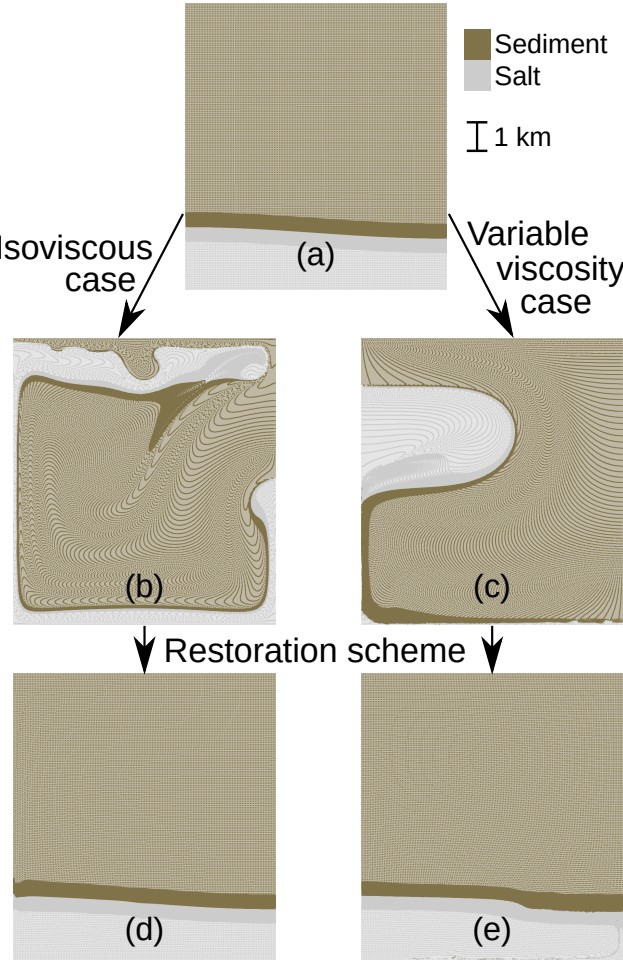

**Figure 5.** Particle swarms for the two synthetic diapiric growth experiments. The darker grey and brown parts on the swarms are due to the higher density of particles around the interfaces. The particles have the same initial position (a) in the two experiments, with different material properties. The result of the forward simulation after $6 \times 10^6$ years for the first experiment is shown in (b). (c) shows the result of the forward simulation for the second experiment after $1.5 \times 10^6$ years. The results for the restoration simulations are shown in (d) and (e) for the first and the second experiment, respectively.

have taken into account the deposition of the sediments at the same time as the formation of the diapir, by removing the sediment layers one by one. For simplification purposes and in order to test the process with simple boundary conditions, such sedimentation processes were not implemented. In this case, the stress state inside the model being incorrect, the sediment and salt layers couldn't be restored to a completely flat state. For example, the shallow sediments should have been removed early in the restoration process, and as such were deformed beyond their sedimentation point. While the results are still quite convincing despite the high level of simplification, this shows that the salt diapir is a result of upbuilding and not downbuilding. The analysis of the five experiments shows that in this setup, the viscosity contrast between salt and sediment and the density of the sediments do not have a big impact on the shape of the model after the restoration process. Only the shape of the sediments at the

base of the diapir is slightly different from one experiment to the other. Experiments 4 and 5 have serrated shapes that are not geologically probable, probably because of the four orders of magnitude of viscosity contrast between the salt and sediments. The main difference between the experiments is the relaxation time for each restoration process. If the duration of the formation of the diapir was known, it could then be used to reduce the uncertainty on which density and viscosity to use. It also seems that the curvature of restored layers changes. This could provide another criterion to further evaluate the results (but would call for variable sediment viscosity testing). However, this is a difficult path forward, because sediment deposition clearly plays a major role during salt displacement (e.g., Giles and Lawton, 1999; Hudec and Jackson, 2007; Rowan et al., 2012). Moreover, recovering the full deformation path during sediment deposition would

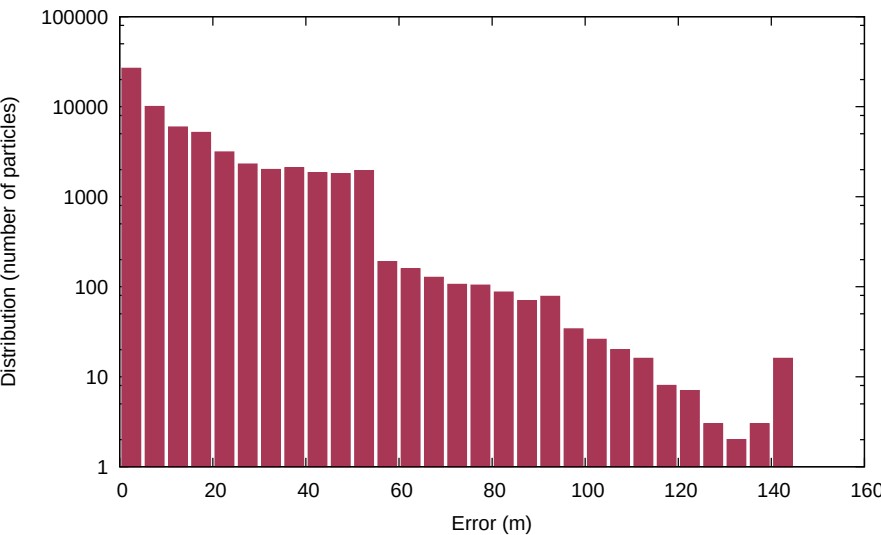

**Figure 6.** Error logarithmic distribution for the first experiment (isoviscosity) on the diapiric growth model.

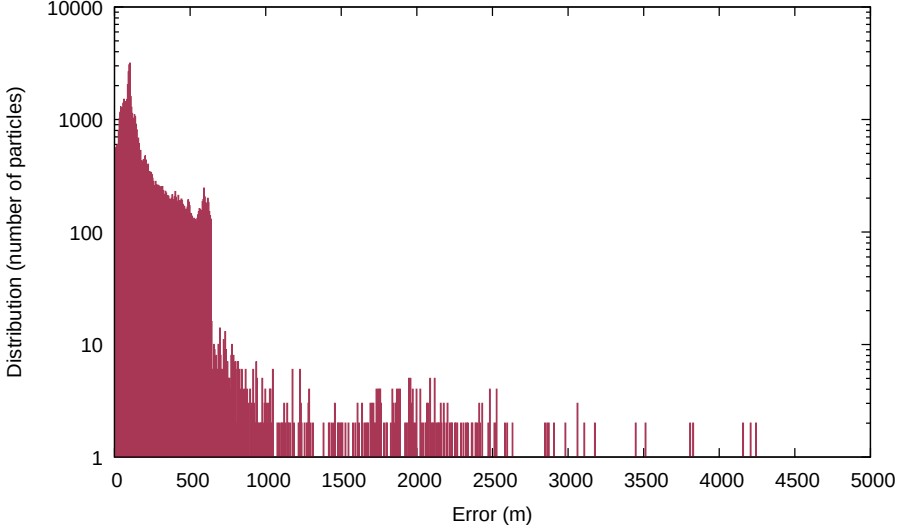

**Figure 7.** Error logarithmic distribution for the second experiment (variable viscosity) on the diapiric growth model.

also call for further studies, possibly on laboratory analogs (Weijermars et al., 1993; Dooley et al., 2005).

### 4.3 Simplified graben model

The last model is a simplification of the creation of a graben in sediments submitted to lateral flow in an extensive context, as shown in Fig. 11.

It consists of a layered overburden underlain by salt and cut by two $60°$ faults. As the intent of the manuscript is to focus on the restoration of structural models, we do not consider the formation of the faults with plasticity, but rather start with two faults already present. The domain size is $6\,\mathrm{km}$ horizontally and $2\,\mathrm{km}$ vertically, and the right boundary is subjected to lateral flow. This is modeled by a Dirichlet condition applying a specific value for the velocity in the horizontal direction (the vertical value for the velocity is still free, as free slip boundary conditions). The left and bottom boundaries are set to free slip, and the top boundary is considered a free surface. For the model to evolve without interference with the lateral flow on the right boundary, the faults are positioned at one third of the model width from the left. In order to capture the geometry of the faults, which is essential in this setup, the adaptive refinement of the grid is an important feature of the proposed implementation. As such, the grid is refined specifically near the faults, with $60 \times 20$ initial elements and four levels of adaptive refinement (up to

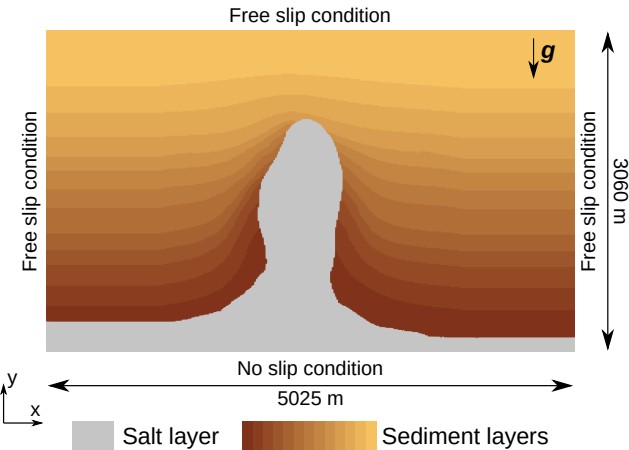

**Figure 8.** Setup of the simulation for the model generated with the method proposed by Clausolles et al. (2019). The initial model is sampled on a regularly spaced particle swarm.

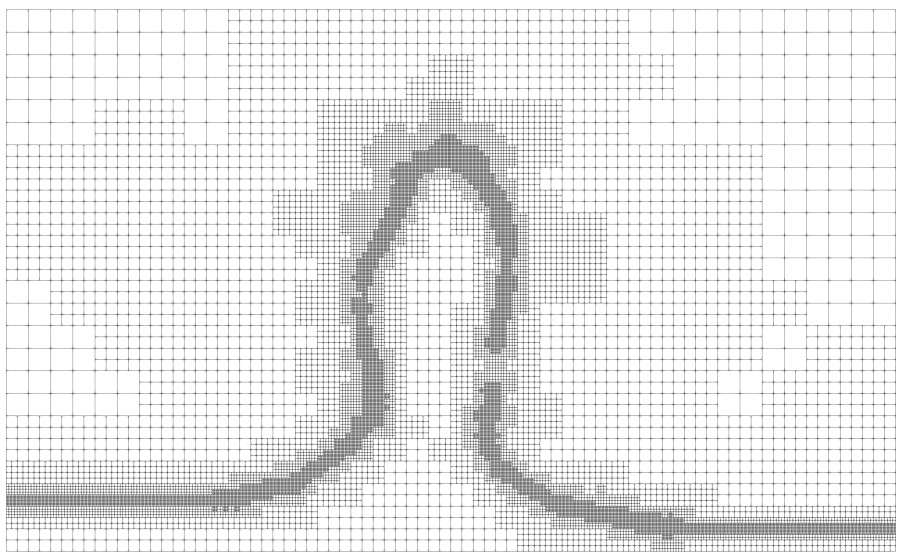

**Figure 9.** Adaptively refined grid for the first time step of the simulation. We can see that the grid is refined to a high level at the interface between the salt and the sediment overburden, where the highest velocity gradients appear. On the contrary, it is coarsened where the velocity has small gradients, particularly in the upper right and upper left corners.

$6.25 \, \text{m} \times 6.25 \, \text{m}$ cells near the faults), as shown in Fig. 12 for the first time step.

The faults are considered as shear bands with a lower viscosity and density than the rest of the overburden. The overburden is layered with two types of rocks with slightly different density and viscosity. Material properties inside the model are:

– Overburden type 1 layer : $\eta_{o1} = 1.5 \times 10^{19}$ Pa.s, $\rho_{o1} = 2550 \, \text{kg.m}^{-3}$

– Overburden type 2 layer : $\eta_{o2} = 5.0 \times 10^{19}$ Pa.s, $\rho_{o2} = 2600 \, \text{kg.m}^{-3}$

– Salt layer : $\eta_s = 1.0 \times 10^{17}$ Pa.s, $\rho_s = 2160 \, \text{kg.m}^{-3}$

– Faults : $\eta_f = 1.0 \times 10^{16}$ Pa.s, $\rho_f = 2200 \, \text{kg.m}^{-3}$

Like in the diapiric growth model, we first do a forward simulation, and then apply the restoration scheme on the model obtained. The lateral flow is set to $10 \, \text{mm/year}$ outwards and the model is observed for $35,000 \, \text{years}$, both in forward and backward simulations, to have sufficient deformation. The particle swarm has a heterogeneous particle density, with a regular sampling in most of the model and eight times more particles near the faults and around the interface between the salt and the overburden. The average distance between two particles in the densified zone is $2.5 \, \text{m}$. The total number of

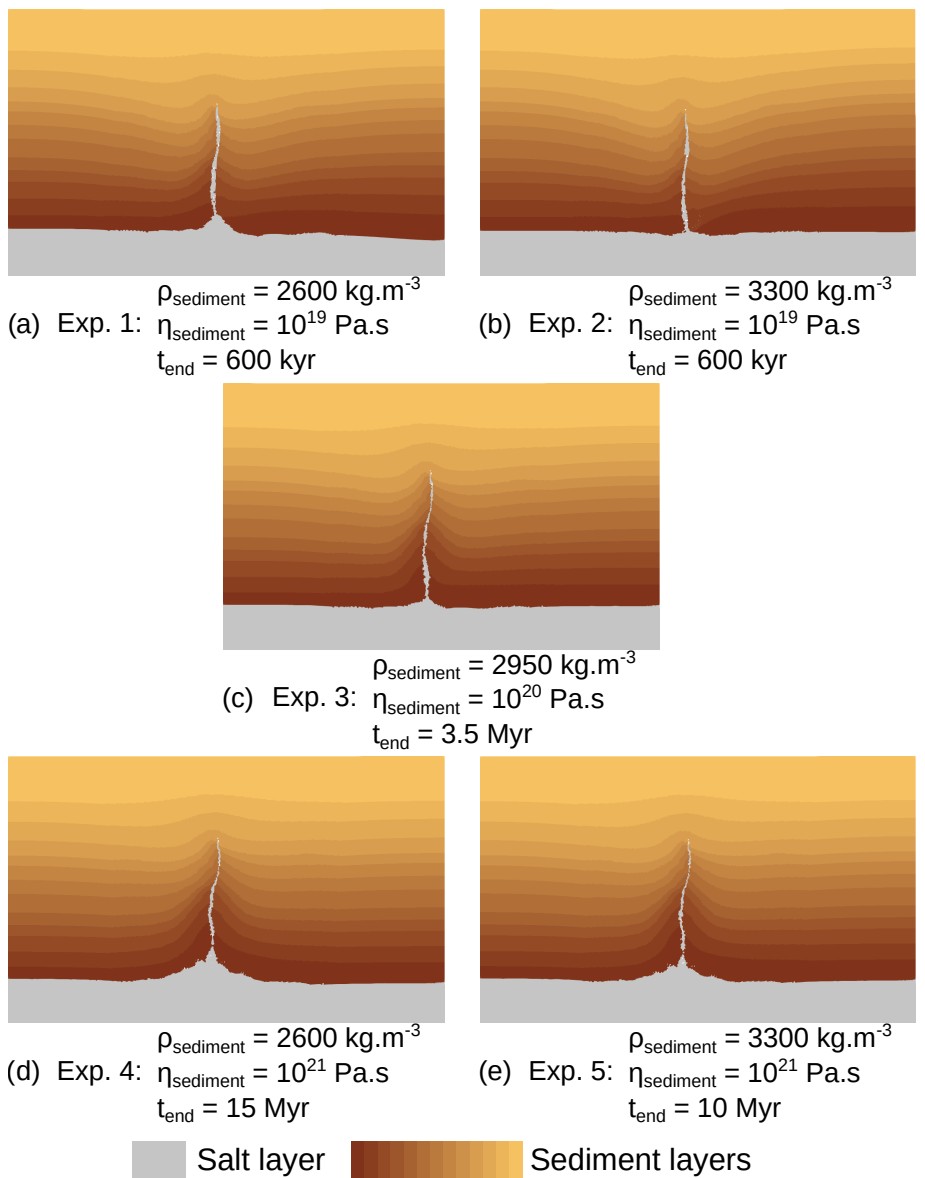

(a)  Exp. 1:  $\rho_{sediment} = 2600$ kg.m$^{-3}$
            $\eta_{sediment} = 10^{19}$ Pa.s
            $t_{end} = 600$ kyr

(b)  Exp. 2:  $\rho_{sediment} = 3300$ kg.m$^{-3}$
            $\eta_{sediment} = 10^{19}$ Pa.s
            $t_{end} = 600$ kyr

(c)  Exp. 3:  $\rho_{sediment} = 2950$ kg.m$^{-3}$
            $\eta_{sediment} = 10^{20}$ Pa.s
            $t_{end} = 3.5$ Myr

(d)  Exp. 4:  $\rho_{sediment} = 2600$ kg.m$^{-3}$
            $\eta_{sediment} = 10^{21}$ Pa.s
            $t_{end} = 15$ Myr

(e)  Exp. 5:  $\rho_{sediment} = 3300$ kg.m$^{-3}$
            $\eta_{sediment} = 10^{21}$ Pa.s
            $t_{end} = 10$ Myr

Salt layer            Sediment layers

**Figure 10.** Results of the 5 restoration experiments done on the salt model setup of Fig. 8, after different time spans $t_{end}$.

particles at the first step of the simulation is about $330,000$. This number decreases during the forward simulation, as the particles are removed once they flow outside of the model. During the restoration simulation, new particles are added near the right boundary when the particle swarm flows inward due to the negative time stepping. Their material properties are determined from the particles moving inwards and their motion. The model at the end of the forward and backward simulation is shown in Fig. 13. Figure 14 shows the difference between the position of the interfaces before the forward simulation and at the end of the backward simulation. The numbering of the interfaces follows their position in the model, interface 0 being the lowest salt-sediment in-

terface. The mean and maximum error for each interface are given in Table 1.

Overall, the results for the restoration simulation are quite good, with mean errors around $1\%$ of the forward deformation for the layer interfaces. Fig. 14 shows that the graben part of the model (between the two faults) is approximatively $7$ m lower than it should be at the end of the restoration. This is due to the model topography being slightly tilted from the right-hand side of the model towards the faults at the end of the restoration. The largest errors are located on the two faults. In particular, the free surface behaves well during the simulation, except for some small instabilities occuring on the top of the faults, where high viscosity contrasts occur. The error resulting from these instabilities, however, is quite

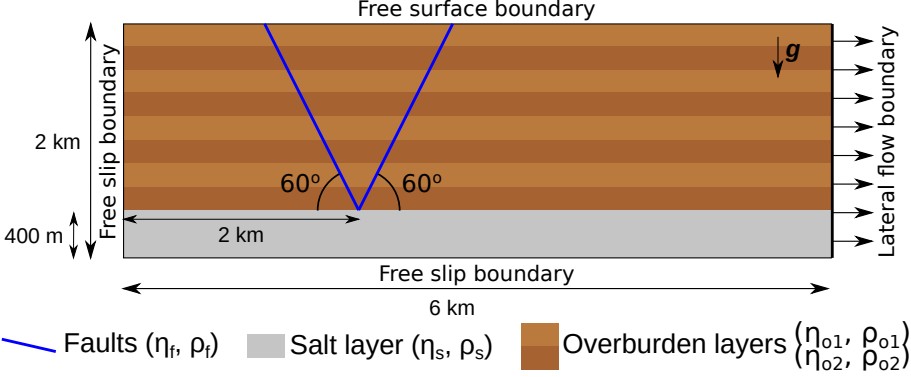

**Figure 11.** Setup of the simulation for the simplified graben model. The initial particle swarm is densified near the faults and at the interface between salt and sediment.

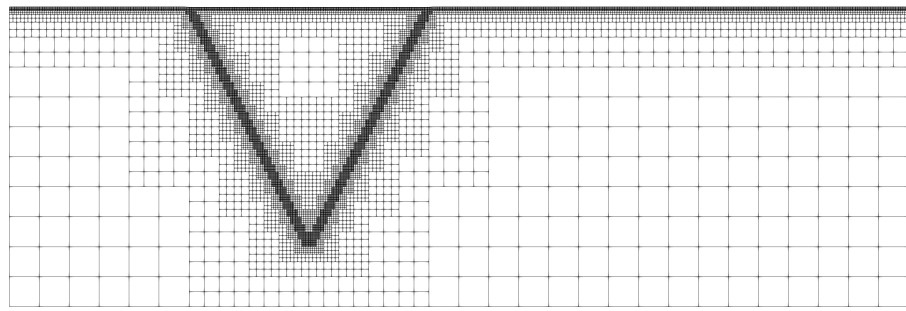

**Figure 12.** Adaptively refined grid for the first time step of the simulation. We can see that the grid is refined to a high level near the faults, where the velocity gradient is high. On the contrary, it is coarsened where the velocity has small gradients, particularly in the lower right and left corners.

small compared to the amount of deformation in the model (around 200 m of slip on the faults).

## 5 Discussion

While the results of the three test models in the previous section are promising, their purpose is not to correctly compute the deformation of the subsurface in a forward mechanical simulation, but rather to assess the validity of the proposed restoration scheme and the underlying concepts in various geological cases.

In this paper, we have made some links between two different types of structural restoration approaches. On one side, geomechanical restoration methods have been relying on considering rocks as elastic material and flattening the top surface of the horizons (e.g., Guzofski et al., 2009; Lovely et al., 2012; Chauvin et al., 2018), and may lead to unphysical strains. On the other side, dynamic restoration methods have considered viscous fluid rheologies for the rock layers, the deformation being driven by density contrasts and boundary conditions, and applied with a backward advection scheme (e.g., Kaus and Podladchikov, 2001; Ismail-Zadeh et al., 2001; Lechmann et al., 2010). As such, restoration us-

ing Stokes equations is expected to provide more physical strains, given that the boundary conditions and the rheology inside the model are close enough to reality. This method is not new (e.g., Kaus and Podladchikov, 2001; Ismail-Zadeh et al., 2001), but has been restricted to the restoration of salt structures and small-scale folds, in environments where the topography remains flat. Here, we are interested in applying it to environments with faults and large displacements of the topography. In this scope, we introduce a numerical scheme combining features of the ALE and PIC approaches, and using adaptive grid refinement. We show that it is accurate enough to produce consistent results on the restoration of models with a viscous backward advection approach.

When applying a reverse-time Stokes restoration scheme, two important questions appear: what are the material properties of the geological objects inside the model, and what type and intensity of boundary conditions should be applied to these geological objects? Regarding the material properties, the diapir test model of Sect. 4.2 gave a first idea of how to choose them, and previous articles have considered the question on specific setups (e.g., Lechmann et al., 2010). The density of the subsurface depends on the type of rocks that are present, and its estimation is relatively easy. The viscosity, however, is not trivial, as laboratory observation time

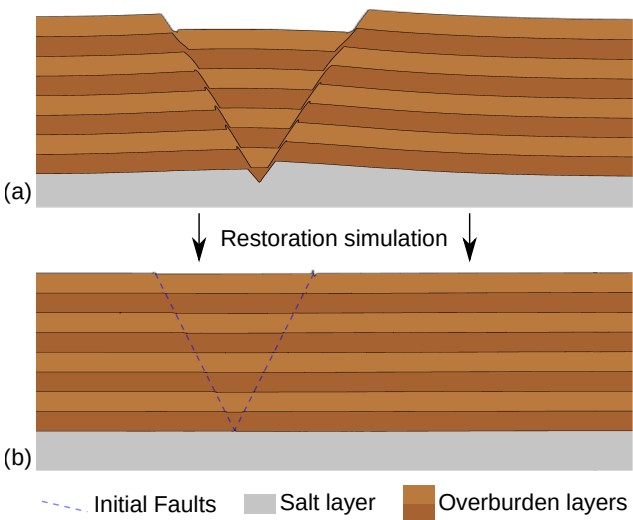

**Figure 13.** Results of the simulations for the simplified graben model at (a) the end of the forward simulation and (b) the end of the restoration simulation.

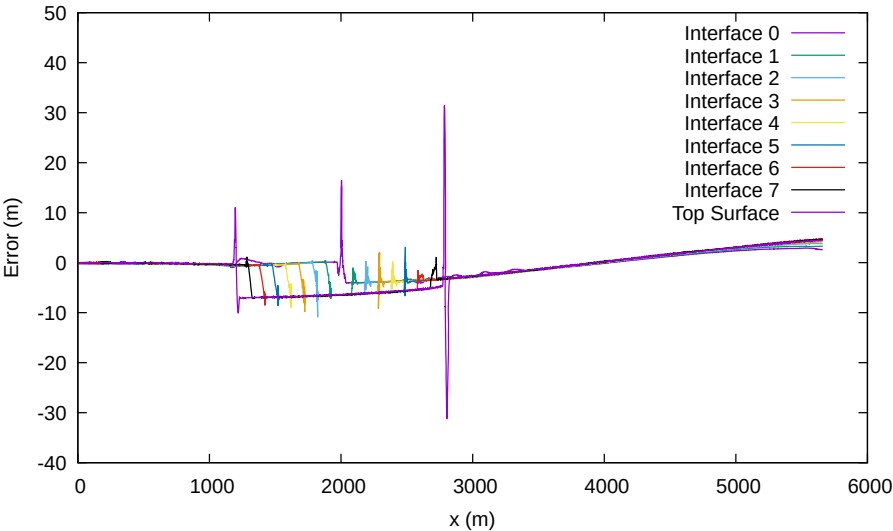

**Figure 14.** Error on the restoration of the interfaces in the model. The lower the number of the interface, the lower the interface they correspond to. For example, interface 0 is the interface between the salt and the lowest sediment layer.

scales are too short to reflect the slow movement occuring at geological time scales. The values we took are inspired from numerical simulations, but they have a large uncertainty (at least one order of magnitude) (e.g., Massimi et al., 2006; Kronbichler et al., 2012), as they are calibrated using postglacial rebound data for example. Works on analog sandbox experiments and further experiments on models with more geological knowledge should prove to be useful in estimating a proper viscosity for the restoration of different rock rheologies. In particular, the duration over which the geologic phenomena occur could guide the choice of viscosity values in subsurface models.

Most geomechanical restoration schemes consider basin rocks to have an elastic behaviour, whereas the rheology used herein does not display Poisson effects. Incorporating elasticity in viscous flow has been done, for example by using an effective viscosity to account for the elastic part of the material while minimizing the modifications to the viscous flow code (e.g., Moresi et al., 2003). The problem is that these schemes, like every implementation of elasticity, use values of the stress and strain at previous time steps. The elastic behavior then depends completely on the stress state at the beginning of the simulation, which is not available in restoration schemes. However, specific material properties could still be taken into account in other ways in Stokes-based restoration.

**Table 1.** Mean and maximum errors for the restoration of each interface.

|             | Mean error (m) | Max error (m) |
|-------------|----------------|---------------|
| Interface 0 | 1.5            | 16.5          |
| Interface 1 | 1.7            | 7.2           |
| Interface 2 | 1.9            | 10.9          |
| Interface 3 | 2.1            | 9.8           |
| Interface 4 | 2.3            | 9.1           |
| Interface 5 | 2.4            | 8.6           |
| Interface 6 | 2.6            | 8.5           |
| Interface 7 | 2.7            | 7.1           |
| Top Surface | 2.9            | 31.4          |

For example, the incompressibility constraint, which implies a Poisson's ratio of 0.5, can be relaxed, which could be used to account for lesser values of the Poisson's ratio. Regarding the rheology of faults, we cannot directly use their usual forward modeling implementation considering the rock as having a plastic behavior. Indeed, the previous stress history is needed to simulate such a behavior, and it is not available in restoration, which studies backward movement. Here, we used a specific viscosity for the implementation of faults in restoration, which holds two advantages. First, since all the faults are already identified at the beginning of the restoration process, we do not need to allow the creation of faults in backward simulations. Second, using an effective viscosity for the faults allows for a more realistic simulation of shear band and damage zone behavior, compared to previous geomechanical restoration schemes that consider faults as free-slip surfaces.

A significant issue with the boundary conditions in geomechanical simulations is the difficulty to estimate the paleo-forces at play several kilometers underground. We therefore need to choose Dirichlet and Neumann boundary conditions that best fit the tectonic knowledge about the region of study. For example, deformation is generally strongly influenced by the horizontal stress state, implying compressive or extensive structures and the need for corresponding conditions on the side boundaries (Chauvin et al., 2018). Another example is the top surface of the models, which can be considered as being on ground level, and is therefore in contact with air. This interface is complicated to handle due to the several orders of magnitude in the material property contrast (very high density and viscosity for rocks versus almost null density and viscosity for the air). In geomechanical simulations, several approaches exist to model its behavior. The simplest topographic surface solution is to set a free-slip condition which removes the normal component of the velocity at the boundary. This simplification is mostly used in cases where the movement of the top surface is negligible compared to the rest of the model. In order to do more realistic simulations, two main approaches are available: the implementation of a free surface, or the "sticky air" method (e.g., Crameri et al., 2012, for a benchmark and a comparison of the two methods). The sticky air method considers a layer of material with a low viscosity and zero density, the difficulty being that this viscosity needs to be sufficiently low to be negligible compared to the rest of the model, but high enough for the solvers to converge. The free surface method considers that no force is applied on the surface of the computational mesh. While this is theoretically simple, it is numerically complicated to implement, as it also means that the computational mesh needs to honor the movement of the free surface. In FAIStokes, the free surface method is applied by tracking the movement of the top surface and allowing the grid nodes to move vertically (Sect. 3.7). In order to stabilize its movement and avoid some of the instabilities that can appear, the free surface stabilization algorithm presented in Kaus et al. (2010) has been implemented. The free surface implementation has been benchmarked and performs well in forward simulations (Appendices D and E). In restoration simulations, the results are more mitigated. Indeed, in models where the only drive is a density contrast (such as the models shown in Sect. 4.1 and Sect. 4.2), the free surface shows instabilities. This appears particularly when working with models that have a near-horizontal or initially horizontal top surface. In those setups, any small computational error in the computation of the vertical part of the velocity can lead to instabilities that increase exponentially in reverse time. Several approaches involving specific tractions on the top surface have been tested to remove or correct this instability, but we have not yet devised any efficient means to prevent it in such setups. In particular, the FSSA delays this phenomenon, but does not suppress it altogether. In models where other drives for the deformation occur, such as lateral flow, the results are more promising, as shown in Sect. 4.3. Indeed, in such setups, the boundary conditions introduce larger strains that dampen the instabilities. For test and comparison purposes, the sticky air method has also been implemented and coupled with the FSSA. It uses the moving grid feature of the free surface so that the cells containing air and rock layers are distinct, and performs similarly to the free surface on the benchmarks presented in Appendices D and E. In restoration simulations, it can delay the instabilities that appear in models driven exclusively by gravity, but doesn't remove it.

## 6  Conclusions

We have presented a scheme that exploits the reversibility
of Stokes equations to perform structural restoration on dif-
ferent geological setups. While the principle of the method
is not new, we have shown that it may be applied on mod-
els with faults and a non-flat topography. As such, it may
improve some of the issues with the current geomechani-
cal restoration implementations that are used for such envi-
ronments. The FAIStokes code was developed to apply this
restoration scheme and allow various tests on its implemen-
tation. Among those tests, we presented three simple mod-
els and the results we obtained with them. Those results are
encouraging, although the numerical method has difficulties
dealing with the restoration of salt in the presence of welds.
The free surface is well managed in our experiment includ-
ing lateral flow, but also leads to instabilities in the restora-
tion process when the flow inside the models is driven ex-
clusively by density. Overall, we still show that combining
adaptive grid refinement with the PIC and ALE approaches
gives enough accuracy to produce consistent restoration re-
sults on different model setups.

We intend to follow this work by applying the method to
more complex models, starting with the restoration of sand-
box experiments (e.g., Colletta et al., 1991). This will allow
us to do more precise tests on the value to choose for the vis-
cosity and density of geological layers, and to upgrade the
specific implementation of faults.

*Code availability.*  The code corresponding to this paper is available
to members of the RING consortium in the FAIStokes software. The
FE parts of the code, however, come from the open source library
deal.II. This library is also used in the open source software AS-
PECT, which also allows the use of PIC and FSSA.

*Video supplement.* For a video example of the restora-
tion of the upscaled van Keken model, viewers can go to
https://doi.org/10.5446/46388. For the restoration of the salt
diapir model, a video of the restoration of Exp. 3 is available at
https://doi.org/10.5446/47889.

## Appendix A:  Taking into account small scales inside a model : the Rayleigh-Taylor instability benchmark

This benchmark is based on the analytical solution of a
Rayleigh-Taylor instability by Ramberg (1968) and was car-
ried out in various numerical studies (Deubelbeiss and Kaus
(2008); Thieulot (2011)). It consists of a two-layer system
driven by gravity, the density of the bottom layer being
smaller. The bottom and top boundaries have a no slip bound-
ary condition, while the sides have a free slip boundary con-
dition.

The first layer, made of fluid 1 with properties $(\rho_1, \eta_1)$,
overlays the second layer, made of fluid 2 $(\rho_2, \eta_2)$. An ini-
tial sinusoidal disturbance of the interface between the two
layers is introduced, characterized by an amplitude $\Delta$ and a
wavelength $\lambda$, as shown in Fig. A1.

Under these conditions, the velocity of the diapiric growth
$v$ is given by Ramberg (1968):

$$\frac{v}{\Delta} = -K \frac{\rho_1 - \rho_2}{2\eta_2} h_2 g \tag{A1}$$

with $K$ the dimensionless growth factor given by

$$K = \frac{-d_{12}}{c_{11}j_{22} - d_{12}i_{21}} \tag{A2}$$

which involves the following factors:

$$\phi_1 = \frac{2\pi h_1}{\lambda}$$

$$\phi_2 = \frac{2\pi h_2}{\lambda}$$

$$c_{11} = \frac{2\eta_1\phi_1^2}{\eta_2(\cosh(2\phi_1) - 1 - 2\phi_1^2)} - \frac{2\phi_2^2}{\cosh(2\phi_2) - 1 - 2\phi_2^2}$$

$$d_{12} = \frac{\eta_1(\sinh(2\phi_1) - 2\phi_1)}{\eta_2(\cosh(2\phi_1) - 1 - 2\phi_1^2)} + \frac{\sinh(2\phi_2) - 2\phi_2}{\cosh(2\phi_2) - 1 - 2\phi_2^2}$$

$$i_{21} = \frac{\eta_1\phi_2(\sinh(2\phi_1) + 2\phi_1)}{\eta_2(\cosh(2\phi_1) - 1 - 2\phi_1^2)} + \frac{\phi_2(\sinh(2\phi_2) + 2\phi_2)}{\cosh(2\phi_2) - 1 - 2\phi_2^2}$$

$$j_{22} = \frac{2\eta_1\phi_1^2\phi_2}{\eta_2(\cosh(2\phi_1) - 1 - 2\phi_1^2)} - \frac{2\phi_2^3}{\cosh(2\phi_2) - 1 - 2\phi_2^2}$$

$$\tag{A3}$$

We set $\rho_1 = 3300$ kg.m$^{-3}$, $\rho_2 = 3000$ kg.m$^{-3}$, $\eta_1 = 10^{21}$ Pa.s, $L_x = h_1 + h_2 = 512$ km, and $\Delta = 3$ km. We make
$\eta_2$ vary between $1.25 \times 10^{20}$ and $2.5 \times 10^{23}$ Pa.s, while $\lambda$
takes three values: $L_x/2, L_x/4, L_x/8$ (Fig. A2).

A first run is done, where the FEM grid is fixed to $80 \times 80$
elements, each containing $10^2$ regularly spaced particles. In
order to test the influence of adaptive refinement, we conduct
a second run with a grid starting at $80 \times 80$ elements and
three levels of adaptive refinement. We also refine the parti-
cle swarm adaptively: each initial cell is first filled with $5^2$
regularly spaced particles, and then the swarm is densified to
64 times more particles around the interface between the two
fluids. The results are shown along with the analytical ones
in Fig. A3.

Overall, results show a good agreement between the com-
puted solution and the reference, especially in the case of
adaptive refinement, where the relative error falls beneath
2.5% for all the curves. Since $\phi_1$ is inversely proportional
to the wavelength $\lambda$, it means that the code can account well
for small disturbances, especially with the use of adaptive re-
finement on the parts with higher velocity and high contrasts
in viscosity.

This benchmark ensures the validity of the code in the
presence of large viscosity constrasts, even if those constrasts

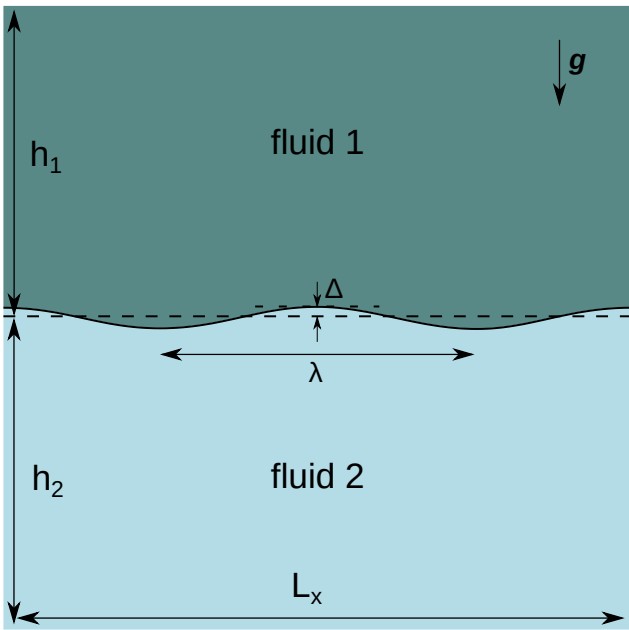

**Figure A1.** Rayleigh-Taylor instability benchmark initial setup.

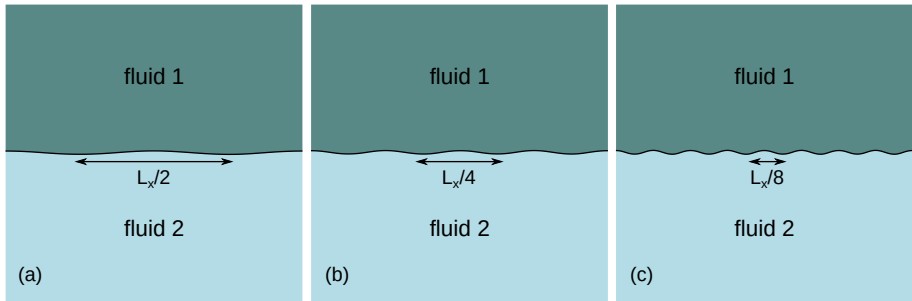

**Figure A2.** Initial setup of the Rayleigh-Taylor instability benchmark with 3 different wavelength: a) $\lambda = L_x/2$, b) $\lambda = L_x/4$, c) $\lambda = L_x/8$

are located on deformations that are small compared to the size of the model. It also validates the averaging of the density and viscosity from the particles to the finite element grid.

## Appendix B: Taking into account viscosity changes : the falling block benchmark

This benchmark appears in Gerya (2019) and is presented in Thieulot (2011). It consists in modelling the fall of a block of fluid of properties $(\rho_1, \eta_1)$ inside another fluid of properties $(\rho_2, \eta_2)$, with $\rho_1 > \rho_2$. The domain is a square of size $L_x = L_y = 500$ km, and the block (a square in 2D) of size $100 \times 100$ km is initially centered at point ($x = 250$ km, $y = 400$ km), as shown in Fig. B1.

The simulation is carried out on a $50 \times 50$ element grid that is adaptively refined three times. Like in the previous benchmark, the particle swarm is created by first introducing $5^2$ particles in each initial element, and then densifying it up to 64 times more particles around the zone of interest (i.e. the falling block). Free slip boundary conditions are imposed on all sides of the domain. We carry out five experiments:

- Exp.1: $\eta_2 = 10^{20}$ Pa.s, $\rho_1 = 3220$ kg.m$^{-3}$;
- Exp.2: $\eta_2 = 10^{21}$ Pa.s, $\rho_1 = 3300$ kg.m$^{-3}$;
- Exp.3: $\eta_2 = 10^{22}$ Pa.s, $\rho_1 = 6600$ kg.m$^{-3}$;
- Exp.4: $\eta_2 = 10^{23}$ Pa.s, $\rho_1 = 3300$ kg.m$^{-3}$;
- Exp.5: $\eta_2 = 10^{24}$ Pa.s, $\rho_1 = 9900$ kg.m$^{-3}$;

In all the experiments, the density of the surrounding fluid is $\rho_2 = 3200$ kg.m$^{-3}$ and the viscosity of the block is varied between $10^{19}$ and $5 \times 10^{27}$ Pa.s. The velocity of the falling block is measured in its centre at $t = 0$ for all experiments. Following physical intuition, one expects the velocity of the block to act as follows: (a) decrease when the viscosity of the surrounding fluid $\eta_2$ increases (i.e. when going

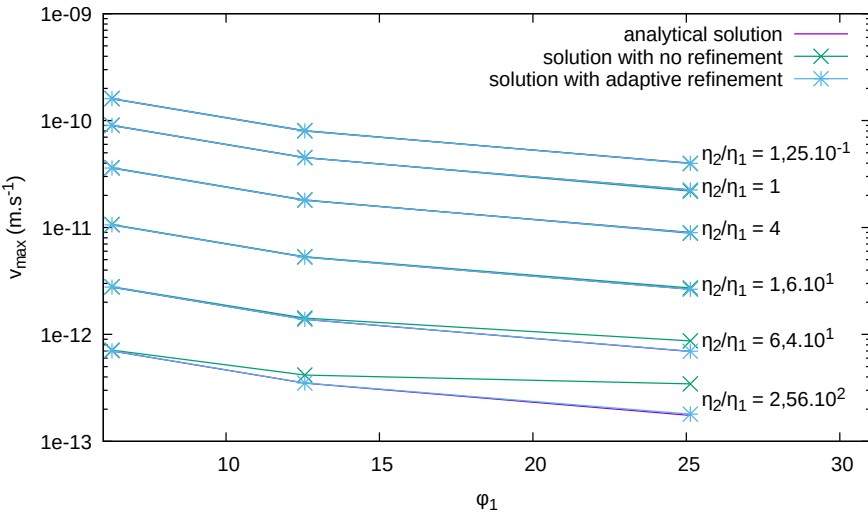

**Figure A3.** Comparison between numerical and analytical results for the Rayleigh-Taylor instability benchmark. The numerical results are computed for a $80 \times 80$ element grid and for the same grid with three levels of adaptive refinement.

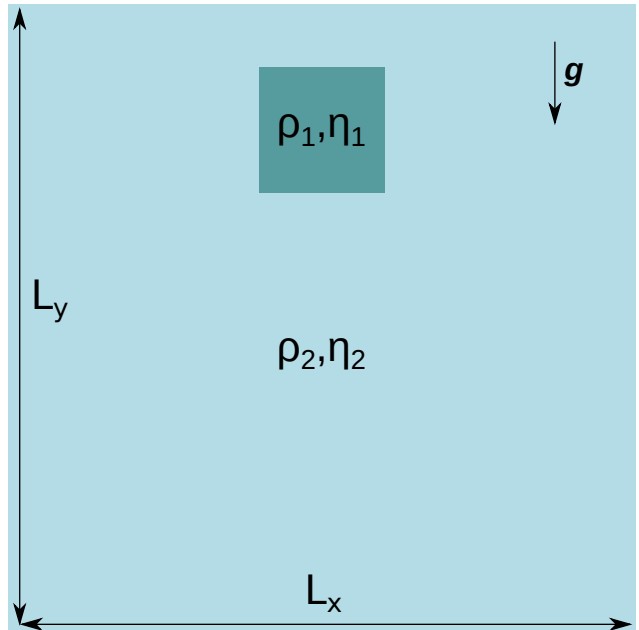

**Figure B1.** Falling block benchmark initial setup.

from Exp.1 to Exp.5), and (b) increase with the density contrast $(\rho_1 - \rho_2)$ in each experiment. To check this behavior, we measure $v\eta_2/(\rho_1 - \rho_2)$ as a function of the viscosity contrast $\log_{10}(\eta_2/\eta_1)$. The results of the benchmark are plotted in Fig. B2.

We can see that the experimental points line up on a single curve, which shows that FAIStokes can deal with gravity-driven simulations where $0.6\% \leq (\rho_1 - \rho_2)/\rho_2 \leq 210\%$ and the viscosity contrasts are as strong as $10^{-6} \leq \eta_2/\eta_1 \leq 10^5$ in a consistent manner.

**Appendix C: Advecting particles : the rotation benchmark**

The last benchmark aims at assessing the error in the advection part only. The setup of the model is a square of size $10 \times 10$ km, where we study the advection of a single particle, starting at coordinates $(8 \text{ km}, 5 \text{ km})$ and doing a $2\pi$ rotation around the center point $(5 \text{ km}, 5 \text{ km})$ (Fig. C1). A velocity field is prescribed in the domain and discretized on the grid: on each grid point, the velocity has a constant norm

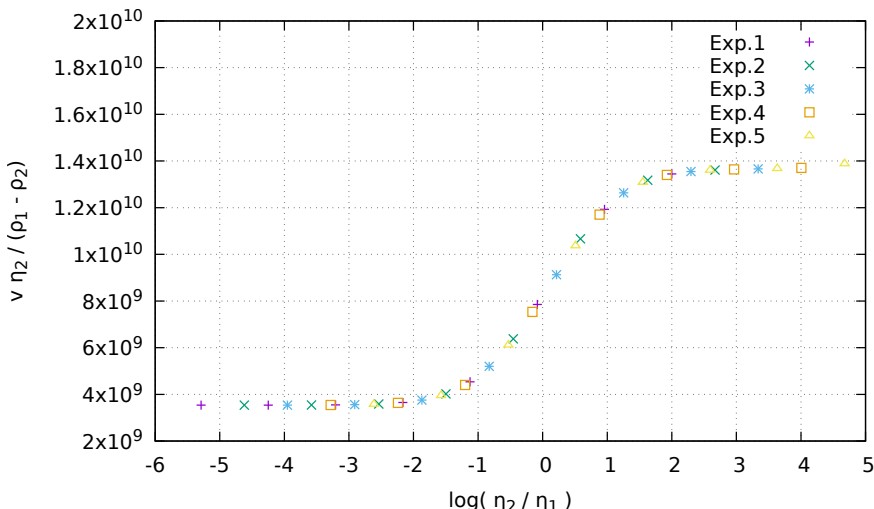

**Figure B2.** Velocity measurements as a function of the viscosity contrast between surrounding medium and block for the experiments of the falling block benchmark.

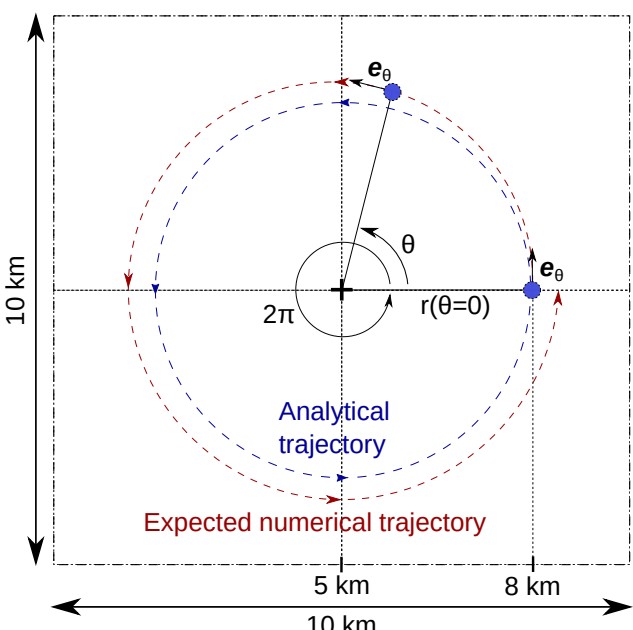

**Figure C1.** Setup for the rotation benchmark, assessing errors on the advection of particles.

and is always normal to the line connecting the point to the model center:

$$\boldsymbol{v} = v_0.\boldsymbol{e_\theta} = \begin{pmatrix} v_0.sin\theta \\ v_0.cos\theta \end{pmatrix} \tag{C1}$$

The grid is not adaptively refined here, and is composed of $16 \times 16$ elements. In order to have scales that are geologically relevant, we choose $v_0 = 3 \, \mathrm{cm.year^{-1}}$ and vary the time step $\Delta t$ between $500$ and $2000$ years (in this setup, the CFL numbers chosen for our simulations would give a timestep be-

tween $175$ years for the lowest CFL number and $1753$ years for the highest CFL number). The second order Runge-Kutta scheme presented in Sect. 3.6 is used at all time steps. We then evaluate the distance $\Delta r = |r(\theta=0) - r(\theta=2\pi)|$. This distance gives us a measure of the error made in the computation of the particle advection, and allow us to compare different advection schemes. Figure C2 shows the results obtained for a $2\pi$ rotation of the particle with different interpolation schemes. We can see that reducing the timestep linearly reduces the error on the radius $r(\theta)$. In this setup, the type of

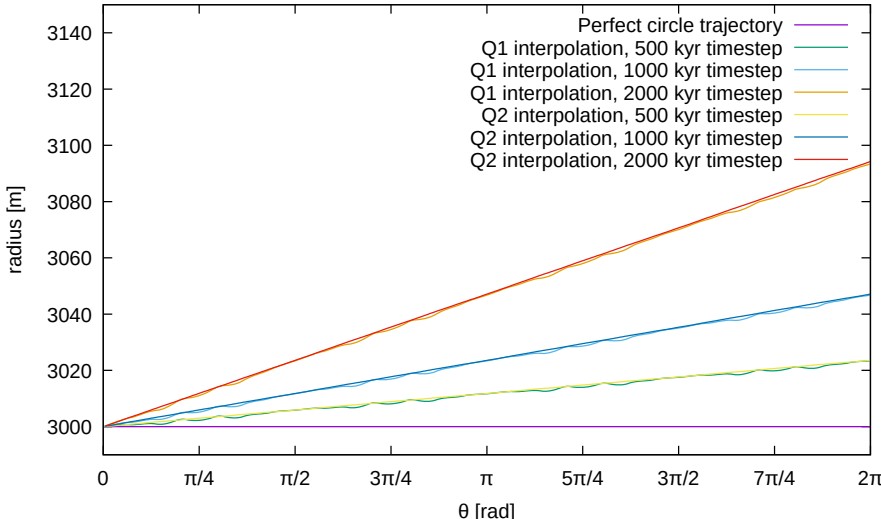

**Figure C2.** Results for the rotation benchmark obtained with different time steps and advection schemes.

interpolation mostly impacts the stability of the interpolation, and not the accuracy.

### Appendix D: Taking into account the top surface in contact with air : the free surface benchmark

This benchmark is presented in Crameri et al. (2012), where it is applied on several numerical codes to compare their implementation of the free surface, and evaluate the use of the 'sticky air' method. It will be used here to evaluate the quality of our approximation and interpolation of the free surface. It consists on a cosine-shaped layer of homogeneous lithosphere overlaying a homogeneous layer of mantle. For this type of model, Ramberg (1981) gives an analytical solution for the maximal height of the topography at each time $t$ :

$$h_{analytical}(t) = h_{initial} \, \exp(-\gamma t) \qquad \text{(D1)}$$

where $\gamma$ is the relaxation rate and $h_{initial}$ is the value of $h$ at the beginning of the simulation. The model setup for the benchmark is shown in Fig. D1.

The bounding box of the model spans $2800 \, \text{km}$ by $707 \, \text{km}$. The underlaying mantle layer is $600 \, \text{km}$ thick, while the lithosphere has a thickness between $93$ and $107 \, \text{km}$. The lithosphere's top surface is cosine-shaped with an amplitude of $7 \, \text{km}$ and a wavelength of the size of the domain. The mantle and lithosphere have a density of $\rho_M = \rho_L = 3300 \, \text{kg.m}^{-3}$ and a viscosity of $\eta_M = 10^{21} \, \text{Pa.s}$ and $\eta_L = 10^{23} \, \text{Pa.s}$, respectively. We set free slip boundary conditions for the sides and a no slip condition on the bottom of the model. The initial grid is made of $16 \times 64$ elements and is adaptively refined 3 times. The particle swarm contains $484,160$ particles; it is constructed by first sampling regularly the domain, and then adaptively densifying the swarm

to 64 times more particles in the lithosphere and upper part of the mantle. In this setup, Crameri et al. (2012) gives a characteristic relaxation rate $\gamma = 0.2139 \times 10^{-11} \, \text{s}^{-1}$ and a characteristic relaxation time $t_{relax} = 14.825 \times 10^3 \, \text{year}$. The results obtained with FAIStokes are given in Fig. D2.

The numerical results are close to the analytical ones, with only a $1.3\%$ error at the characteristic relaxation time. This shows the capacity of FAIStokes to compute the solution of Stokes equations with a free surface for small vertical deformation, and to advect the particles inside the model. It also gives another evaluation of the handling of gravity-driven flow, this time with the addition of evolution inside the model.

### Appendix E: Upgrading the free surface movement : the sloshing benchmark

This benchmark is presented in Kaus et al. (2010), where it is used to assess the results of the free surface stabilization algorithm (FSSA) presented in the same article. It is used here to verify the implementation of the same algorithm in our code, as well as check the behavior of the free surface in another setup. The benchmark model is another Rayleigh-Taylor instability with a dense, more viscous layer sinking into a less dense fluid (Fig. E1).

The model span is $500 \, \text{km} \times 500 \, \text{km}$; the side boundaries have a free-slip condition, the lower boundary is no-slip, and the top boundary is a free surface. The initial perturbation between the two layers is sinusoidal with an amplitude of $5 \, \text{km}$. The computation is carried out on a grid with $25 \times 25$ initial elements and three adaptive refinement steps. The particle swarm counts $25,000$ particles; it is constructed by first sampling regularly the model and then densifying it

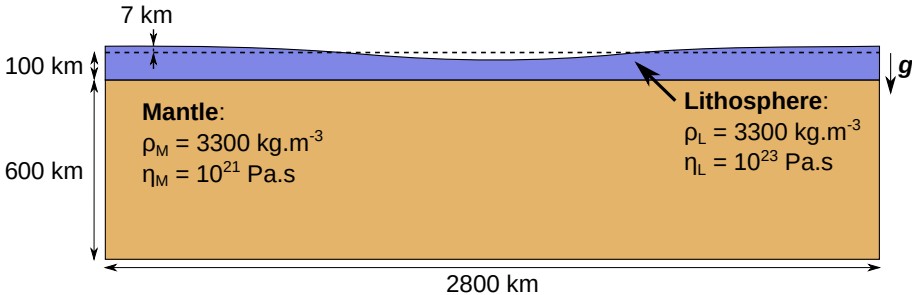

**Figure D1.** Model setup for the 2D free surface benchmark.

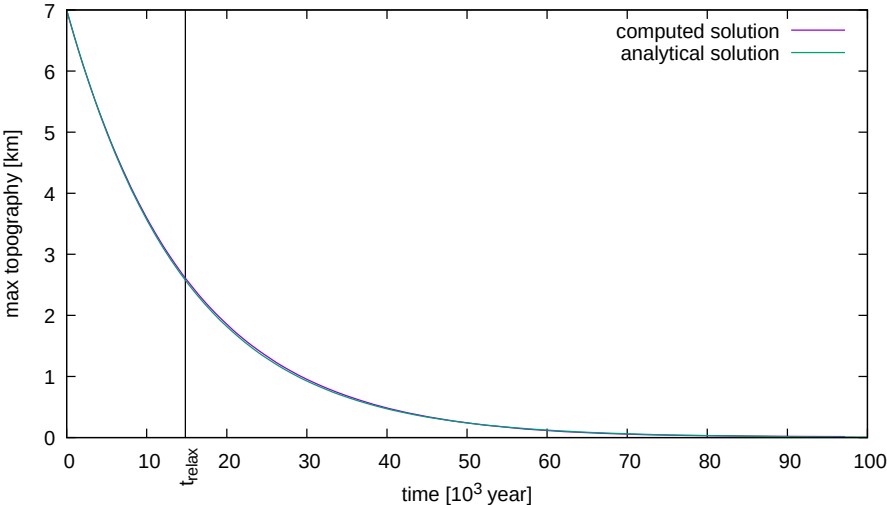

**Figure D2.** Comparison between the analytical and numerical results of the maximum topography over time for the free surface benchmark.

to 64 times more particles around the interface. The specificity of this benchmark is the apparition of a sloshing instability (also refered to as the "drunken sailor" instability) if the simulation time step is too large. Specifically here, without the FSSA, the forward simulation is stable with a time step $\Delta t$ of 2500 years, but with $\Delta t = 5000$ years, an instability emerges as the velocity pattern changes direction from one time step to the other (Fig E2).

In order to follow the evolution of the free surface, we keep trace of the altitude of the most top-left point over time. Results of a 0.5 Myr simulation, for different time steps $\Delta t$, with and without the FSSA, are shown in Fig. E3. We can see that the implementation of the FSSA stabilizes the sloshing behavior of the free surface that appeared whith a time step $\Delta t$ of 5000 years, and keeps the free surface stable even with higher time steps. Moreover, the results show great similarities to those that can be found in Kaus et al. (2010) and Thieulot (2019). This validates the implementation of the free surface stabilization algorithm. It also gives another evaluation of the handling of gravity-driven flow with a free surface, this time with the additional resolution of an instability that can occur with free surfaces.

*Author contributions.* The writing of this paper, as well as the code of the software and the results, was done by M S-S. CT helped with useful discussions and new points of view on the subject of the paper, as well as help on the code structure, benchmarks, and methods. GC and PC provided useful discussions on the subject, guidance on solving the different problems, and funding for the PhD thesis that allowed this research through the RING-Gocad consortium. All authors contributed to the conceptualization, the design of experiments and the analysis of results. CT, GC and PC also helped on the manuscript with useful feedback and corrections.

*Competing interests.* The authors declare that they have no conflict of interests.

*Acknowledgements.* This project was done in the frame of the RING project, in the GeoRessources laboratory. We would like to thank for their support the academic and industrial sponsors of the RING-GOCAD Consortium managed by ASGA (https://www.ring-team.org/consortium). We would also like to thank Jean Braun for valuable discussions and suggestions on this work.

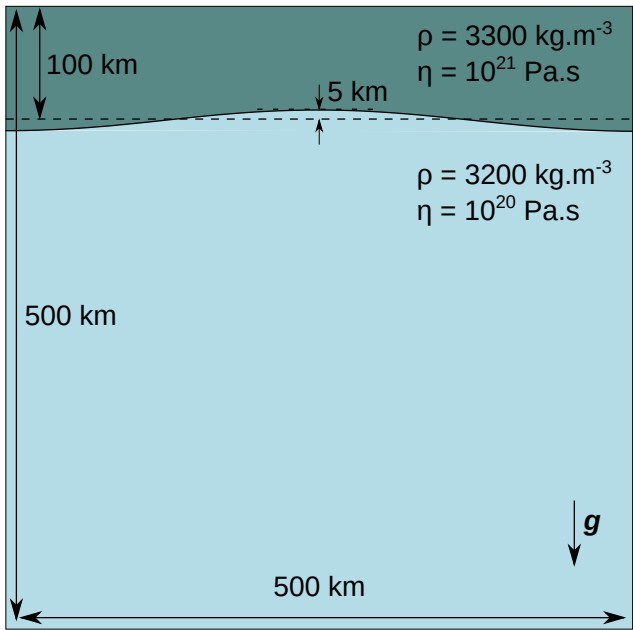

**Figure E1.** Sloshing free surface benchmark initial setup.

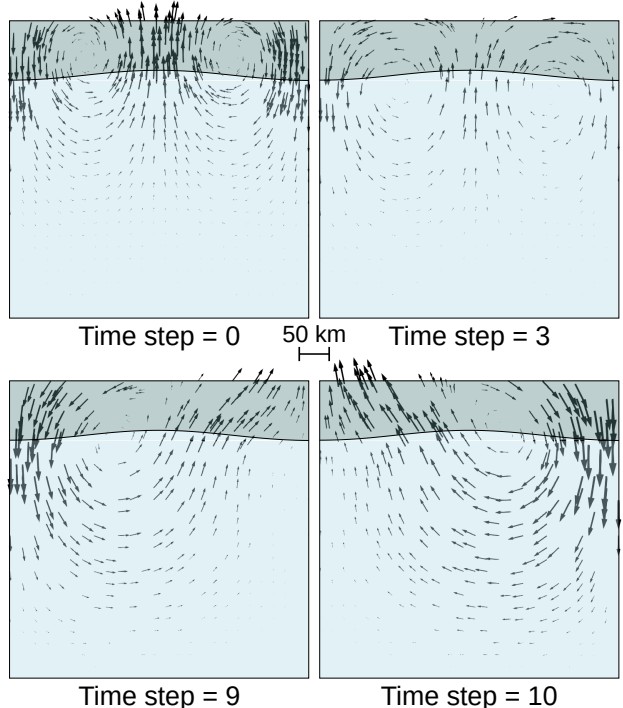

**Figure E2.** Simulation evolution for $\Delta t = 5000$ years, showing the sloshing instability: the velocity pattern changes from one time step to the other, the velocity norm increasing each time.

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

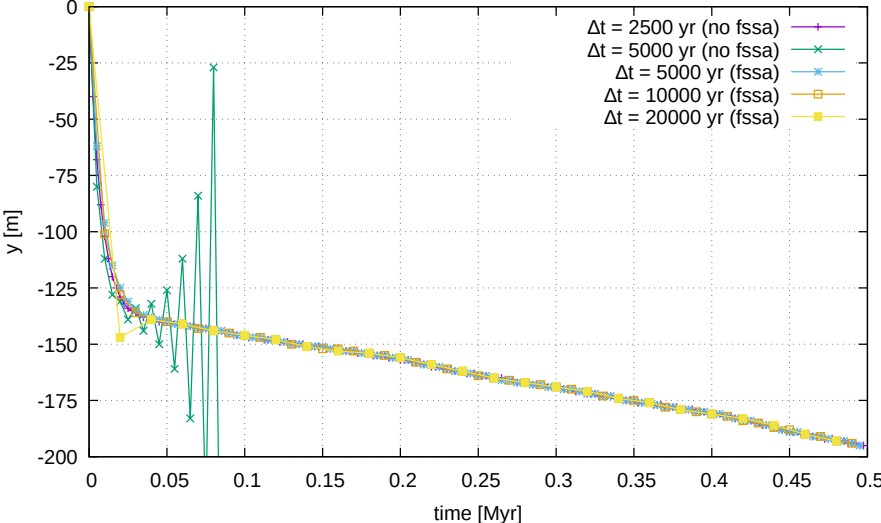

**Figure E3.** Altitude of the most top-left point of the grid over time, for the sloshing free surface benchmark, for diffent time steps with and without the FSSA.

ical maps and cross-sections for numerical simulations, Comptes Rendus Geoscience, 351, 48–58, 2019.

Arndt, D., Bangerth, W., Clevenger, T. C., Davydov, D., Fehling, M., Garcia-Sanchez, D., Harper, G., Heister, T., Heltai, L., Kronbichler, M., Kynch, R. M., Maier, M., Pelteret, J.-P., Turcksin, B., and Wells, D.: The deal.II Library, Version 9.1, Journal of Numerical Mathematics, https://doi.org/10.1515/jnma-2019-0064, https://dealii.org/deal91-preprint.pdf, accepted, 2019.

Arndt, D., Bangerth, W., Davydov, D., Heister, T., Heltai, L., Kronbichler, M., Maier, M., Pelteret, J.-P., Turcksin, B., and Wells, D.: The deal. II finite element library: Design, features, and insights, Computers & Mathematics with Applications, https://doi.org/https://doi.org/10.1016/j.camwa.2020.02.022, 2020.

Asgari, A. and Moresi, L.: Multiscale Particle-In-Cell Method: From Fluid to Solid Mechanics, in: Advanced Methods for Practical Applications in Fluid Mechanics, edited by Jones, S. A., chap. 9, IntechOpen, Rijeka, https://doi.org/10.5772/26419, https://doi.org/10.5772/26419, 2012.

Athy, L. F.: Density, porosity, and compaction of sedimentary rocks, AAPG Bulletin, 14, 1–24, 1930.

Bangerth, W., Hartmann, R., and Kanschat, G.: deal.II – a General Purpose Object Oriented Finite Element Library, ACM Trans. Math. Softw., 33, 24/1–24/27, 2007.

Bouziat, A., Guy, N., Frey, J., Colombo, D., Colin, P., Cacas-Stentz, M.-C., and Cornu, T.: An Assessment of Stress States in Passive Margin Sediments: Iterative Hydro-Mechanical Simulations on Basin Models and Implications for Rock Failure Predictions, Geosciences, 9, 469, 2019.

Braun, J.: Pecube: A new finite-element code to solve the 3D heat transport equation including the effects of a time-varying, finite amplitude surface topography, Computers & Geosciences, 29, 787–794, 2003.

Chamberlin, R. T.: The Appalachian folds of central Pennsylvania, The Journal of Geology, 18, 228–251, 1910.

Chauvin, B. P., Lovely, P. J., Stockmeyer, J. M., Plesch, A., Caumon, G., and Shaw, J. H.: Validating novel boundary conditions for three-dimensional mechanics-based restoration: An extensional sandbox model example, AAPG Bulletin, 102, 245–266, 2018.

Clausolles, N., Collon, P., and Caumon, G.: Generating variable shapes of salt geobodies from seismic images and prior geological knowledge, Interpretation, 7, T829–T841, 2019.

Cobbold, P. R. and Percevault, M.-N.: Spatial integration of strains using finite elements, in: Strain Patterns in Rocks, pp. 299–305, Elsevier, 1983.

Colletta, B., Letouzey, J., Pinedo, R., Ballard, J. F., and Balé, P.: Computerized X-ray tomography analysis of sandbox models: Examples of thin-skinned thrust systems, Geology, 19, 1063–1067, 1991.

Cornet, F. H.: Elements of crustal geomechanics, Cambridge University Press, 2015.

Courant, R., Friedrichs, K., and Lewy, H.: Über die partiellen Differenzengleichungen der mathematischen Physik, Mathematische annalen, 100, 32–74, 1928.

Crameri, F., Schmeling, H., Golabek, G., Duretz, T., Orendt, R., Buiter, S., May, D., Kaus, B., Gerya, T., and Tackley, P.: A comparison of numerical surface topography calculations in geodynamic modelling: an evaluation of the 'sticky air' method, Geophysical Journal International, 189, 38–54, 2012.

Dahlstrom, C.: Balanced cross sections, Canadian Journal of Earth Sciences, 6, 743–757, 1969.

De Santi, M. R., Campos, J. L. E., and Martha, L. F.: A Finite Element approach for geological section reconstruction, in: Proceedings of the 22th Gocad Meeting, Nancy, France, pp. 1–13, Citeseer, 2002.

Deubelbeiss, Y. and Kaus, B.: Comparison of Eulerian and Lagrangian numerical techniques for the Stokes equations in the presence of strongly varying viscosity, Physics of the Earth and Planetary Interiors, 171, 92–111, 2008.

Dimakis, P., Braathen, B. I., Faleide, J. I., Elverhøi, A., and Gud-laugsson, S. T.: Cenozoic erosion and the preglacial uplift of the Svalbard–Barents Sea region, Tectonophysics, 300, 311–327, 1998.

Donea, J., Huerta, A., Ponthot, J.-P., and Rodriguez-Ferran, A.: Arbitrary Lagrangian-Eulerian Methods, volume 1 of Encyclopedia of Computational Mechanics, chapter 14, John Wiley & Sons Ltd, 3, 1–25, 2004.

Dooley, T., McClay, K., Hempton, M., and Smit, D.: Salt tectonics above complex basement extensional fault systems: results from analogue modelling, in: Geological Society, London, Petroleum Geology Conference series, vol. 6, pp. 1631–1648, Geological Society of London, 2005.

Durand-Riard, P., Caumon, G., and Muron, P.: Balanced restoration of geological volumes with relaxed meshing constraints, Computers & Geosciences, 36, 441–452, 2010.

Durand-Riard, P., Salles, L., Ford, M., Caumon, G., and Pellerin, J.: Understanding the evolution of syn-depositional folds: Coupling decompaction and 3D sequential restoration, Marine and Petroleum Geology, 28, 1530–1539, 2011.

Durand-Riard, P., Guzofski, C., Caumon, G., and Titeux, M.-O.: Handling natural complexity in three-dimensional geomechanical restoration, with application to the recent evolution of the outer fold and thrust belt, deep-water Niger Delta, AAPG bulletin, 97, 87–102, 2013a.

Durand-Riard, P., Shaw, J. H., Plesch, A., and Lufadeju, G.: Enabling 3D geomechanical restoration of strike-and oblique-slip faults using geological constraints, with applications to the deep-water Niger Delta, Journal of Structural Geology, 48, 33–44, 2013b.

Fernandez Terrones, N.: 2D and 3D numerical modelling of multi-layer detachment folding and salt tectonics, Ph.D. thesis, Mainz University.

Fillon, C., Huismans, R. S., and van der Beek, P.: Syntectonic sedimentation effects on the growth of fold-and-thrust belts, Geology, 41, 83–86, https://doi.org/10.1130/G33531.1, 2013.

Fletcher, R. C. and Pollard, D. D.: Can we understand structural and tectonic processes and their products without appeal to a complete mechanics?, Journal of Structural Geology, 21, 1071–1088, 1999.

Fossen, H.: Structural geology, Cambridge University Press, 2016.

Fullsack, P.: An arbitrary Lagrangian-Eulerian formulation for creeping flows and its application in tectonic models, Geophysical Journal International, 120, 1–23, 1995.

Gassmöller, R., Heien, E., Puckett, E. G., and Bangerth, W.: Flexible and scalable particle-in-cell methods for massively parallel computations, arXiv preprint arXiv:1612.03369, 2016.

Gassmöller, R., Lokavarapu, H., Heien, E., Puckett, E. G., and Bangerth, W.: Flexible and Scalable Particle-in-Cell Methods With Adaptive Mesh Refinement for Geodynamic Computations, Geochemistry, Geophysics, Geosystems, 19, 3596–3604, https://doi.org/10.1029/2018GC007508, 2018.

Gassmöller, R., Lokavarapu, H., Bangerth, W., and Puckett, E. G.: Evaluating the accuracy of hybrid finite element/particle-in-cell methods for modelling incompressible Stokes flow, Geophysical Journal International, 219, 1915–1938, https://doi.org/10.1093/gji/ggz405, 2019.

Gerbault, M., Poliakov, A. N., and Daignieres, M.: Prediction of faulting from the theories of elasticity and plasticity: what are the limits?, Journal of Structural Geology, 20, 301–320, 1998.

Gerya, T.: Introduction to numerical geodynamic modelling, Cambridge University Press, 2019.

Giles, K. A. and Lawton, T. F.: Attributes and evolution of an exhumed salt weld, La Popa basin, northeastern Mexico, Geology, 27, 323–326, 1999.

Gratier, J.-P.: L'équilibrage des coupes géologiques. Buts, méthodes et applications., Géosciences-Rennes, 1988.

Groshong, R.: 3-D structural geology, Springer, 2006.

Guzofski, C. A., Mueller, J. P., Shaw, J. H., Muron, P., Medwedeff, D. A., Bilotti, F., and Rivero, C.: Insights into the mechanisms of fault-related folding provided by volumetric structural restorations using spatially varying mechanical constraints, AAPG bulletin, 93, 479–502, 2009.

Heister, T., Dannberg, J., Gassmöller, R., and Bangerth, W.: High accuracy mantle convection simulation through modern numerical methods–II: realistic models and problems, Geophysical Journal International, 210, 833–851, https://doi.org/10.1093/gji/ggx195, 2017.

Hudec, M. R. and Jackson, M. P.: Terra infirma: Understanding salt tectonics, Earth-Science Reviews, 82, 1–28, 2007.

Hughes, T. J.: The finite element method: linear static and dynamic finite element analysis, Courier Corporation, 2012.

Ismail-Zadeh, A. and Tackley, P.: Computational methods for geodynamics, Cambridge University Press, 2010.

Ismail-Zadeh, A., Tsepelev, I., Talbot, C., and Korotkii, A.: Three-dimensional forward and backward modelling of diapirism: numerical approach and its applicability to the evolution of salt structures in the Pricaspian basin, Tectonophysics, 387, 81–103, 2004.

Ismail-Zadeh, A. T., Talbot, C. J., and Volozh, Y. A.: Dynamic restoration of profiles across diapiric salt structures: numerical approach and its applications, Tectonophysics, 337, 23–38, 2001.

Kaus, B. J. and Podladchikov, Y. Y.: Forward and reverse modeling of the three-dimensional viscous Rayleigh-Taylor instability, Geophysical Research Letters, 28, 1095–1098, 2001.

Kaus, B. J., Mühlhaus, H., and May, D. A.: A stabilization algorithm for geodynamic numerical simulations with a free surface, Physics of the Earth and Planetary Interiors, 181, 12–20, 2010.

Kocher, T. and Mancktelow, N. S.: Dynamic reverse modelling of flanking structures: a source of quantitative kinematic information, Journal of Structural Geology, 27, 1346–1354, 2005.

Koyi, H.: Salt flow by aggrading and prograding overburdens, Geological Society, London, Special Publications, 100, 243–258, 1996.

Kronbichler, M., Heister, T., and Bangerth, W.: High accuracy mantle convection simulation through modern numerical methods, Geophysical Journal International, 191, 12–29, 2012.

Lechmann, S. M., Schmalholz, S. M., Burg, J.-P., and Marques, F.: Dynamic unfolding of multilayers: 2D numerical approach and application to turbidites in SW Portugal, Tectonophysics, 494, 64–74, 2010.

Lovely, P., Flodin, E., Guzofski, C., Maerten, F., and Pollard, D. D.: Pitfalls among the promises of mechanics-based restoration: Addressing implications of unphysical boundary conditions, Journal of Structural Geology, 41, 47–63, 2012.

Lovely, P. J., Jayr, S. N., and Medwedeff, D. A.: Practical and efficient three-dimensional structural restoration using an adaptation of the GeoChron model, AAPG Bulletin, 102, 1985–2016, 2018.

Maerten, F. and Maerten, L.: Unfolding and Restoring Complex Geological Structures Using Linear Elasticity Theory, in: AGU Fall Meeting Abstracts, 2001.

Maerten, L. and Maerten, F.: Chronologic modeling of faulted and fractured reservoirs using geomechanically based restoration: Technique and industry applications, AAPG bulletin, 90, 1201–1226, 2006.

Massimi, P., Quarteroni, A., and Scrofani, G.: An adaptive finite element method for modeling salt diapirism, Mathematical Models and Methods in Applied Sciences, 16, 587–614, 2006.

Massimi, P., Quarteroni, A., Saleri, F., and Scrofani, G.: Modeling of salt tectonics, Computer methods in applied mechanics and engineering, 197, 281–293, 2007.

Massot, J.: Implémentation de méthodes de restauration équilibrée 3D, Ph.D. thesis, Institut National Polytechnique de Lorraine, 2002.

Moresi, L., Dufour, F., and Mühlhaus, H.-B.: A Lagrangian integration point finite element method for large deformation modeling of viscoelastic geomaterials, Journal of Computational Physics, 184, 476–497, 2003.

Moretti, I.: Working in complex areas: New restoration workflow based on quality control, 2D and 3D restorations, Marine and Petroleum Geology, 25, 205–218, 2008.

Moretti, I., Lepage, F., and Guiton, M.: KINE3D: a new 3D restoration method based on a mixed approach linking geometry and geomechanics, Oil & Gas Science and Technology, 61, 277–289, 2006.

Muron, P.: Méthodes numériques 3-D de restauration des structures géologiques faillées, Ph.D. thesis, INPL, 2005.

Parquer, M. N., Collon, P., and Caumon, G.: Reconstruction of Channelized Systems Through a Conditioned Reverse Migration Method, Mathematical Geosciences, 49, 965–994, 2017.

Pellerin, J., Lévy, B., Caumon, G., and Botella, A.: Automatic surface remeshing of 3D structural models at specified resolution: A method based on Voronoi diagrams, Computers & Geosciences, 62, 103–116, 2014.

Poliakov, A. N., Podladchikov, Y. Y., Dawson, E. C., and Talbot, C. J.: Salt diapirism with simultaneous brittle faulting and viscous flow, Geological Society, London, Special Publications, 100, 291–302, 1996.

Quinquis, M. E., Buiter, S. J., and Ellis, S.: The role of boundary conditions in numerical models of subduction zone dynamics, Tectonophysics, 497, 57–70, 2011.

Ramberg, H.: Instability of layered systems in the field of gravity., Physics of the Earth and Planetary Interiors, 1, 427–447, 1968.

Ramberg, H.: Gravity, deformation and the earth's crust: in theory, experiments and geological application, Academic press, 1981.

Ramón, M. J., Pueyo, E. L., Caumon, G., and Briz, J. L.: Parametric unfolding of flexural folds using palaeomagnetic vectors, Geological Society, London, Special Publications, 425, 247–258, 2016.

Rose, I., Buffett, B., and Heister, T.: Stability and accuracy of free surface time integration in viscous flows, Physics of the Earth and Planetary Interiors, 262, 90–100, 2017.

Rouby, D.: Restauration en carte des domaines faillés en extension. Méthode et applications., Ph.D. thesis, Université Rennes 1, 1994.

Rowan, M. G., Lawton, T. F., and Giles, K. A.: Anatomy of an exposed vertical salt weld and flanking strata, La Popa Basin, Mexico, Geological Society, London, Special Publications, 363, 33–57, 2012.

Royden, L. and Keen, C.: Rifting process and thermal evolution of the continental margin of eastern Canada determined from subsidence curves, Earth and Planetary Science Letters, 51, 343–361, 1980.

Schmalholz, S. M.: 3D numerical modeling of forward folding and reverse unfolding of a viscous single-layer: Implications for the formation of folds and fold patterns, Tectonophysics, 446, 31–41, 2008.

Tang, P., Wang, C., and Dai, X.: A majorized Newton-CG augmented Lagrangian-based finite element method for 3D restoration of geological models, Computers & Geosciences, 89, 200–206, 2016.

Thielmann, M., May, D., and Kaus, B.: Discretization errors in the hybrid finite element particle-in-cell method, Pure and Applied Geophysics, 171, 2165–2184, 2014.

Thieulot, C.: FANTOM: Two-and three-dimensional numerical modelling of creeping flows for the solution of geological problems, Physics of the Earth and Planetary Interiors, 188, 47–68, 2011.

Thieulot, C., Steer, P., and Huismans, R.: Three-dimensional numerical simulations of crustal systems undergoing orogeny and subjected to surface processes, Geochemistry, Geophysics, Geosystems, 15, 4936–4957, 2014.

Thieulot, C. C.: Fieldstone: The Finite Element Method in Computational Geodynamics, https://doi.org/10.23644/uu.9209393.v1, https://uu.figshare.com/articles/manual_pdf/9209393, 2019.

Trim, S., Lowman, J., and Butler, S.: Improving mass conservation with the tracer ratio method: application to thermochemical mantle flows, Geochemistry, Geophysics, Geosystems, 2019.

van Keken, P., King, S., Schmeling, H., Christensen, U., Neumeister, D., and Doin, M.-P.: A comparison of methods for the modeling of thermochemical convection, Journal of Geophysical Research: Solid Earth, 102, 22 477–22 495, 1997.

Weijermars, R., Jackson, M. t., and Vendeville, B.: Rheological and tectonic modeling of salt provinces, Tectonophysics, 217, 143–174, 1993.

Willett, S., Beaumont, C., and Fullsack, P.: Mechanical model for the tectonics of doubly vergent compressional orogens, Geology, 21, 371–374, 1993.

Zehner, B., Hellwig, O., Linke, M., Görz, I., and Buske, S.: Rasterizing geological models for parallel finite difference simulation using seismic simulation as an example, Computers & geosciences, 86, 83–91, 2016.