# Peer review of "Towards the application of Stokes flow equations to structural restoration simulations"

_Solid Earth, 2020_

## Referee Comment (RC1) · Frantz Maerten (Referee) · 27 May 2020

The article presents in a clear concise manner a new way of doing structural restoration using Stokes flow equations. The manuscript is well written and reads smoothly. The use of Stokes flow is clearly justified by the authors in the light of the geomechanical restoration problems (e.g., the non-physical constraint of flattening) but also when considering the difficulty of restoring structures with salt intrusions.

I like publications that are based on simple ideas (here, the *reverse time scheme* used by the authors): ***Everything should be made as simple as possible, but not simpler***. I think that the authors are paving the way for new ideas and developments in the domain of structural restoration, and we clearly see the potential for restoring more

and more complex models, not only in 2D but also in 3D.

Some questions, suggestions:

- Even if faults are not yet included in the modeling, I do not see potential problems as the authors already deal with salt intrusion (interface between the rocks and the salt body). A specific viscosity for the faults can be used for the modeling, which was stated by the authors. So my first question is why the authors did not present a (synthetic) model with at least one faults, as all the ingredients are already here (coding)?

- My second question (and suggestion) is related to rock properties, especially the poisson's ratio and the Young modulus. Is there a way to incorporate those properties in the process of restoration using Stoke flow equations? I think that this problem should be a little bit discuss by the authors as they can have an impact on the restoration process.

- Another suggestion is to provide information about the computation time of the models (or at least for some of them).

Overall, I would say it is an excellent manuscript.

Frantz

---

## Referee Comment (RC2) · Peter Lovely (Referee) · 25 Jun 2020

"Towards the application of Stokes flow equations to structural restoration simulations" presents a novel approach to structural restoration based upon principles of Stokes flow and deformation of Newtonian viscous fluids. The manuscript is well written and organized. The authors clearly explain the new approach and its implementation, provide clear and sound justification for the scientific principles, and demonstrate its potential value with three simple synthetic examples. While the current implementation and demonstration is limited to 2D, the potential extension to 3D is made clear. The manuscript is clearly worthy of publication, but I would first provide several comments and recommendations.

[Figure]

First off, it is my opinion that the authors do not adequately address their assumption of a linear (Newtonian) viscosity model in sections 1 & 2. The authors explain at some length in the introduction the limitations of elastic geomechanical restoration techniques to capture inelastic (nonlinear) processes. They also provide references to justify the representation of rock deformation as viscous flow. However, there is only brief mention in the discussion (line 297) of their simplification to assume Newtonian fluids. At the least, this assumption, and that most of the preferred representations of rock deformation as viscous fluids assume non-Newtonian (e.g. power law) models, needs additional (and earlier) acknowledgement and discussion. The first two examples of the new restoration technique use forward models that also assume Newtonian fluids. These are insightful; however, ideally, I would like to see a restoration of a forward model that uses more realistic rheology for the forward model.

Second, I believe that the third example (Section 4.3) requires additional explanation and discussion.

1. It's not clear how the geometry was constructed. To what extent is this "image" an interpretation of real data vs. based on a model? How was it generated? The general reader should not have to read the reference to understand this model which is critical to this manuscript.

2. Also, why use a stochastically generated diapir rather than a previously published interpretation of a real structure? As a geologist, I would be more comfortable with an example that used a real subsurface structure than a stochastic model.

Further, the results of this section are very interesting, and probably warrant additional discussion.

1. It makes sense that the system tends toward a state that is in mechanical equilibrium (thus a flat salt-sediment interface). It would be nice to know that the restoration path is valid, too.

2. I'm having trouble understanding what are the geologic implications of the restored images in which synkinematic sediments are not removed. What is or is not representative of past state? What parts of the restored images should I focus on (and what should I not focus on)? There are significant differences between the models in the shallower section, but perhaps the authors do not discuss because they consider it geologically irrelevant.

3. It is important to note that the loading of shallower (younger) sediment is not removed and thus the stress state driving restoration in the past is incorrect.

4. A video (or several key frames) of the preferred restoration as it progresses back in time might add value in addition to showing only the final state of each.

The authors discuss the ability of this method to discuss faulted structures, and it seems the numerical implementation is ready. It would be nice to include an example.

The discussion of the numerical implementation (Section 3) is lengthy, and this detracts from the focus on structural restoration. Further, there are many prior implementations of Stokes flow using particle-in-cell methods. I recognize that the numerical implementation was much of the effort, but consider if it would be appropriate to condense this section and move the details to the Appendices (along with the validation examples). This could provide space in the manuscript for additional examples and discussion.

Finally, following are a few more technically specific comments. Figure 1: Verify that the velocity fields (B&C) correspond to this sketch (A). I think that these velocity fields represent a single wavelength perturbation of the material contrast in the horizontal dimension, but the sketch shows two wavelengths perturbation. In other words, for this sketch, there should be four convection cells, not two, and material should be flowing up at the side boundaries in the forward sense.

Paragraph beginning line 279: Use of the term "weld" in the sense of restoration is

confusing. The diapir is restored, and sediment is juxtaposed against sediment where there was originally no salt. This is not a weld in the geologic sense. To avoid confusion, I would recommend finding an alternate description of this feature of restoration.

Paragraph beginning line 321: The authors provide two solutions to the rock-air (or -water) interface problem: sticky air or the free surface. They go on to explain the issues with a free surface in some detail, but do not offer further discussion of the sticky-air solution. If it is a viable solution, why not demonstrate it?

Reference to Medwedeff., et al.2016 (abstract) is now available in peer reviewed paper (Lovely, Jayr & Medwedeff, AAPG Bulletin, 2018)

Lines 65-67: I don't understand why large deformation and potential remeshing may limit the value for interpretation validation. Would remeshing not be OK, so long as key structural elements (e.g. faults and horizons) are preserved?

Line 111: Should a reference be provided for CFL condition?

Lines 136-139: Another reason not to solve the thermal equations is that diffusion may be important at geologic time scales, and it is not reversible.

———————————————

---

## Short Comment (SC1) · 29 Jun 2020

I have read the manuscript of Schuh-Senlis et al. with great interest, in which the authors propose a *"new approach for restoration based on considering geological materials as highly viscous quasi-static fluids"* (line 6-7).

Developing dynamic restoration techniques is an important topic that may indeed guide structural reconstructions and thus improve simple kinematic restorations. As such, this has been a research topic over the past few decades. Whereas many industry-codes rely on assuming a purely elastic overburden rheology, this does not work well in areas governed by viscous flow. Therefore, alternative approaches have discussed doing this by assuming the rheology to be (nonlinear) viscous.

[Figure]

To my surprise, however, none of the previous literature on this topic is cited or properly discussed here. The impression that you give in the manuscript that you discuss a 'new' method is clearly incorrect; performing restoration by making the timestep negative and the rheology viscous has been proposed earlier and in multiple papers – even in the context of the Rayleigh-Taylor instability.

It might be that 'old' becomes 'new' after 10-15 years, or that you simply overlooked this in your literature search. Therefore, here a summary of some of the previous work that I believe to be relevant in this context. Links are given to the publications.

*Kaus Podladchikov (2001) Forward and reverse modelling of the three-dimensional Rayleigh-Taylor instability. Geophysical Research Letters, Vol. 28 (6), p.1095-1098.*
https://doi.org/10.1029/2000GL011789
In this paper, we discussed 3D models of the Rayleigh-Taylor instability, and show that an initial 2D perturbation becomes unstable and breaks up into 3D structures. Importantly, in the same paper (Fig. 4) we also show that one can start from these complex-looking 3D structures and model the structure backwards in time by making the timestep negative to retrieve the initial 2D perturbation (something that is certainly not obvious from looking at the 3D patterns). This paper was limited to iso-viscous cases and was applied to a synthetic case rather than to a natural application.

*Ismail-Zadeh, Talbot, and Volozh. (2001). Dynamic Restoration of Profiles across Diapiric Salt Structures: Numerical Approach and Its Applications. Tectonophysics 337, p. 23-38.*
https://doi.org/10.1016/S0040-1951(01)00111-1
In this paper, the dynamic restoration method is used for viscous materials in the context of salt tectonics, both for synthetic examples with no slip upper boundary conditions and with erosion/deposition/free-surface conditions (hence very similar to the current manuscript). The models were 2D, but took linear (variable) viscosity into account as well as a depth-dependent density structure. In addition to synthetic examples, they also applied the method to a natural case study in the Pricaspian basin.

*Ismail-Zadeh, Tsepelev, Talbot and Korotkii (2004). Three-dimensional forward and backward modelling of diapirism: numerical approach and its applicability to the evolution of salt structures in the Pricaspian basin. Tectonophysics 387 p. 81-103.*
https://doi.org/10.1016/j.tecto.2004.06.006
Whereas you do cite this paper in your current manuscript, and thus likely read it, you only mention it in the context of 3D forward models and boundary conditions for salt tectonics. Yet, as the 2001 paper of the same group, it discusses a dynamic restoration method using viscous rheologies and negative timesteps, but this time in 3D (for Newtonian, variable, viscosities).

*Lechmann, Schmalholz, Burg and Marques (2010). Dynamic unfolding of multilayers: 2D numerical approach and application to turbidites in SW Portugal. Tectonophysics 494 (1-2), p. 64-74.*
https://doi.org/10.1016/j.tecto.2010.08.009
In this paper, the authors demonstrate that the dynamic restoration method also works for cases with a nonlinear (power-law) viscosities, by performing forward and reverse simulations of multilayer stack that produces folds. Using synthetic simulations, the authors demonstrate that it is only possible to retrieve flat layers for the correct viscosity pre-factor and power-law exponents. A subsequent application to a natural case study shows that it mostly works, apart from at a specific location within the folded stack where fieldwork determine that there was a significant amount of out-of-plane flow.

*Kocher and Mancktelow (2005): Dynamic reverse modelling of flanking structures: a source of quantitative kinematic information. Journal of Structural Geology 27 (2005) 1346–1354*
https://doi.org/10.1016/j.jsg.2005.05.007
This paper is on a slightly different scale, but employs the same time-reverse approach to study the bending of layers around a pre-existing weak zone. Here an analytical solution is employed of a thin ellipse and the authors show that such reverse modelling approach combined with field information gives information about both the amount of

strain as well as about the boundary condition that were active.

*Naiara Fernandez (2014). 2D and 3D numerical modelling of multilayer detachment folding and salt tectonics. PhD thesis, Uni Mainz.*
https://dspace-dev.ub.uni-mainz.de/handle/20.500.12030/148446
In chapter 5 of her PhD thesis, which can be downloaded from the address listed above, Naiara Fernandez demonstrates that time-reverse structural restoration method works for salt tectonics with a powerlaw overburden (with n=5), in 2D numerical simulations, and for fully 3D cases with sedimentation/erosion. If the correct parameters are employed, a (nearly) flat salt layer can be recovered whereas clear artifacts occur when, for example, a wrong overburden viscosity is employed.

As the (non-exhaustive) list above thus demonstrates, there is a quite rich literature in dynamic restoration of geological structures using essentially the same or very similar methods to what you discuss in your manuscript. In fact, some papers already studied topics that you mention as being important to address in future research (sedimentation, nonlinear rheologies, 3D). What can perhaps be considered a new contribution in your work is that you show that the methodology works in 2D using adaptive mesh refinement methods, and that you apply it to cases where the free surface does not remain flat.

It is in my opinion part of good scientific conduct to properly discuss and acknowledge previous work and I therefore hope that you will modify the revision version of your manuscript accordingly.

―――――――――――――――――――――

---

## Short Comment (SC2) · 2 Jul 2020

Thank you for this comment.

Indeed, this part of the literature was overlooked, as the possibility of restoring salt structures through viscous rheology in geological models was not mentioned in the articles on structural restoration I read, except in Ismail-Zadeh et al. (2004), that I misread at the time, not realizing the method was already used for restoration purposes.

I would like to apologize for this overlook, that I assure you was not voluntary. I am currently working on revising the manuscript to ensure the previous works on this subject are duly acknowledged. On that subject, I am eager to complete and read this part of the literature and have already started doing so, but I wasn't able to download Naiara

Fernandez's PhD thesis from the link you provided. Perhaps the domain cannot be accessed from outside the University ? If that is the case, could you please send me the PhD manuscript so that I may read it ?

---

## Short Comment (SC3) · 3 Jul 2020

I have read the manuscript submitted to the journal with an interest refreshing my memory of research done some 20+ years ago and to see what new is done in this area. Unfortunately, I did not find any novelty of the work, except an implementation of different numerical solution approach to compute a viscous flow. I respect the work done, and believe that the authors did not know about the work previously done by others (my colleague Boris Kaus mentioned a number of earlier works on this topic). Meanwhile without assessment of the previous work versus of the work presented in this manuscript, the value of the paper will be little, if any. More detailed comments follow.

[Figure]

1. The authors should discuss the novelty point of the work. The statements as "We have therefore developed a new approach for restoration based on considering ... (in the abstract)", "we investigate a new method to address these challenges ... (in the Introduction)", "We have presented a new scheme that exploits ... (in the conclusion)", and elsewhere in the manuscript must be deleted as the approach is NOT new.

The idea of the reconstruction method was presented at the EGS General Assembly in Hague in 1999 and published in 2000 in the Russian book series "Computational Seismology and Geodynamics" and later peer-reviewed and published by AGU https://agupubs.onlinelibrary.wiley.com/doi/abs/10.1002/9781118669853.ch4

2. Careful reading of the manuscript has revealed that the only novelty is the use of different numerical method to calculate the viscous flow (the use of FAIStokes code based on ALE implementation). The authors can highlight this, but not the novel idea of restoration.

Please note that you refer wrongly (line 151) to the technique, which was used in Ismail-Zadeh et al. (2004): the Eulerian FEM was used with tricubic splines to approximate basis functions (see Appendices A-E in Ismail-Zadeh et al., 2004; ).

Also, note (lines 147-148) that the phrase "... neither of them is specifically adapted in the case of large displacements over time" needs clarification. Please see Naimark and Ismail-Zadeh (GJI, 1995) and Naimark, Ismail-Zadeh and Jacoby (GJI, 1998) for the techniques on how to track the interface between the layers and to get a higher accuracy at larger displacements.

3. The Stokes equations.

- Lines 79-80. "(isothermal)" should be deleted as unrelated to the Stokes equations. Also, the Stokes equations are not reversible (see next comments) but the Navier-Stokes equations, which are not used here.

- Lines 89-103. Your creeping flow equations do not depend on time and describe a

steady-state flow. You cannot reverse time in Eqs. (1)-(6) as time does not exist in the equations. To avoid the problem you need to advect density and viscosity with the flow using d\rho/dt=0 and d\eta/dt (d is the full derivative). The fact that FAIStokes treats with the issue using a particle swarm is not enough to ignore the discussion of why the steady-state equations (1)-(6) allow for a non-stationary flow. You must clarify this.

- Lines 105-119. Suddenly you introduce time using the Euler scheme, which is proven to be computationally unstable, and the use of higher-order methods should be encouraged unless your time step is small enough (see e.g., Ismail-Zadeh and Tackley, 2010. Computational Methods for Geodynamics, Cambridge Univ. Press). BTW, later it is mentioned that the second-order RK-method is used (better the 4th-order methods to see a difference in computations).

4. Unstable simulations (lines 127-9). Instabilities in this case reflect inaccuracy of numerical methods used and are likely to be related to overshoots and undershoots of the viscosity, which lead to computation of erroneous velocities controlling the advection of interfaces between layers (including a free surface, where the viscosity ratio is significant). For more detail, please see Naimark, Ismail-Zadeh and Jacoby (1998) - fig. 3 and description in section "Efficiency of the method" (http://www.mitp.ru/∼aismail/papers/GJI1998.pdf)

5. All models of salt diapirism considered in the manuscript are related to up-building (when a lower density salt penetrated into already formed overburden due to RT instability). Actually, in salt tectonics upbuilding is a rare process, and essentially salt structures are formed due to downbuilding (when salt starts to move due to different loading of sedimentary overburden). See examples of upbuilt and downbuit diapir's dynamic restorations in Ismail-Zadeh et al. (2001; http://www.mitp.ru/∼aismail/papers/restore.pdf). This should be discussed in the paper and especially in the light of how the method used can treat the case of downbuilding. A model of downbuilt diapir dynamic restoration using the FAIStokes will enhance the paper.

[Figure]

Hope the comments would help in revising the manuscript.

Alik Ismail-Zadeh

---

## Short Comment (SC4) · 8 Jul 2020

Apologies, the link to the PhD thesis of Naiara Fernandez I gave is indeed only accessible from our University network. Yet, you can find a copy of her thesis on ResearchGate as well:

https://www.researchgate.net/publication/342666004_2D_and_3D_numerical_
modelling_of_multilayer_detachment_folding_and_salt_tectonics

---

## Author Comment (AC1) · 4 Aug 2020

The article presents in a clear concise manner a new way of doing structural restoration using Stokes flow equations. The manuscript is well written and reads smoothly. The use of Stokes flow is clearly justified by the authors in the light of the geomechanical restoration problems (e.g., the non-physical constraint of flattening) but also when considering the difficulty of restoring structures with salt intrusions. I like publications that are based on simple ideas (here, the reverse time scheme used by the authors): Everything should be made as simple as possible, but not simpler. I think that the authors are paving the way for new ideas and developments in the domain of structural restoration, and we clearly see the potential for restoring more and more complex models, not only

[Figure]

**in 2D but also in 3D.**

Thank you for the very positive review and constructive feedback. Following are answers to your questions. The manuscript has also been changed according to those questions. In the following, for an easier reading, the reviewer comments are set in bold, and the answers follow in normal script.

• **Even if faults are not yet included in the modeling, I do not see potential problems as the authors already deal with salt intrusion (interface between the rocks and the salt body). A specific viscosity for the faults can be used for the modeling, which was stated by the authors. So my first question is why the authors did not present a (synthetic) model with at least one faults, as all the ingredients are already here (coding)?**

\* After considering the reviews and commentaries on the manuscript, in order to add more value and explain further the possible applications of the restoration idea, another simple example of a model containing two faults and a free surface was added and discussed in the manuscript.

• **My second question (and suggestion) is related to rock properties, especially the poisson's ratio and the Young modulus. Is there a way to incorporate those properties in the process of restoration using Stoke flow equations? I think that this problem should be a little bit discuss by the authors as they can have an impact on the restoration process.**

\* Incorporating elasticity in viscous flow has been done, for example by using an effective viscosity to account for the elastic part of the material while minimizing the modifications to the viscous flow code (e.g. Moresi et al., 2003). The problem is that those schemes, like every implementation of elasticity, use values of the stress and strain at previous time steps. This poses the problem of the stress state at the begining of the simulation, on which the elastic behaviour part depends completely and which is not available in restoration schemes. However, specific material properties

could still be taken into account in other ways in the restoration process. For example, the incompressibility constraint, which implies a Poisson's ratio of 0.5, can be relaxed (e.g. Thieulot, 2011 for the relaxation of the incompressibility), which could be used to account for lesser values of the Poisson's ratio. This discussion has also been added to the paper, in the discussion section.

• **Another suggestion is to provide information about the computation time of the models (or at least for some of them).**

\* The computation time was not added since the point is not set on code efficiency, and the code has not been parallelized, but it can be done.
* * *

---

## Author Comment (AC2) · 4 Aug 2020

We thank the reviewers F.Maerten and P.Lovely for the time invested in the review, as well as their positive commentaries and constructive feedback. We also thank B.Kaus and A.Ismail-Zadeh for taking the time to read and comment on the manuscript. We apologize for overlooking some of their previous work, which shed another light on the novelty of the work done. The manuscript was modified according to those reviews. In particular, in order to go further with the method and add novelty to the work, the results of simulations on a new model including faults and a non-flat free surface were added and discussed in the manuscript. Here follow answers to the questions of the reviewers (for an easier reading, the reviewer comments are set in italics, and the answers follow in normal script).

[Figure]

**Reviewer 1 (Frantz Maerten):**

*The article presents in a clear concise manner a new way of doing structural restoration using Stokes flow equations. The manuscript is well written and reads smoothly. The use of Stokes flow is clearly justified by the authors in the light of the geomechanical restoration problems (e.g., the non-physical constraint of flattening) but also when considering the difficulty of restoring structures with salt intrusions. I like publications that are based on simple ideas (here, the reverse time scheme used by the authors): Everything should be made as simple as possible, but not simpler. I think that the authors are paving the way for new ideas and developments in the domain of structural restoration, and we clearly see the potential for restoring more and more complex models, not only in 2D but also in 3D.*

\* *Even if faults are not yet included in the modeling, I do not see potential problems as the authors already deal with salt intrusion (interface between the rocks and the salt body). A specific viscosity for the faults can be used for the modeling, which was stated by the authors. So my first question is why the authors did not present a (synthetic) model with at least one faults, as all the ingredients are already here (coding)?*

After considering the reviews and commentaries on the manuscript, in order to add more value and explain further the possible applications of the restoration idea, another simple example of a model containing two faults and a free surface was added and discussed in the manuscript.

\* *My second question (and suggestion) is related to rock properties, especially the poisson's ratio and the Young modulus. Is there a way to incorporate those properties in the process of restoration using Stoke flow equations? I think that this problem should be a little bit discuss by the authors as they can have an impact on the restoration process.*

Incorporating elasticity in viscous flow has been done, for example by using an effective viscosity to account for the elastic part of the material while minimizing the modifications to the viscous flow code (e.g. Moresi et al., 2003). The problem is that those schemes, like every implementation of elasticity, use values of the stress and strain at previous time steps. This poses the problem of the stress state at the begining of the simulation, on which the elastic behaviour part depends completely and which is not available in restoration schemes. However, specific material properties could still be taken into account in other ways in the restoration process. For example, the incompressibility constraint, which implies a Poisson's ratio of 0.5, can be relaxed (e.g. Thieulot, 2011 for the relaxation of the incompressibility), which could be used to account for lesser values of the Poisson's ratio. This discussion has also been added to the paper, in the discussion section.

*\* Another suggestion is to provide information about the computation time of the models (or at least for some of them).*

The computation time was not added since the point is not set on code efficiency, and the code has not been parallelized, but it can be done.

**Reviewer 1 (Frantz Maerten):**

*"Towards the application of Stokes flow equations to structural restoration simulations" presents a novel approach to structural restoration based upon principles of Stokes flow and deformation of Newtonian viscous fluids. The manuscript is well written and organized. The authors clearly explain the new approach and its implementation, provide clear and sound justification for the scientific principles, and demonstrate its potential value with three simple synthetic examples. While the current implementation and demonstration is limited to 2D, the potential extension to 3D is made clear. The manuscript is clearly worthy of publication, but I would first provide several comments and recommendations.*

*\* First off, it is my opinion that the authors do not adequately address their assumption of a linear (Newtonian) viscosity model in sections 1  2. The authors explain at some length in the introduction the limitations of elastic geomechanical restoration*

*techniques to capture inelastic (nonlinear) processes. They also provide references to justify the representation of rock deformation as viscous flow. However, there is only brief mention in the discussion (line 297) of their simplification to assume Newtonian fluids. At the least, this assumption, and that most of the preferred representations of rock deformation as viscous fluids assume non-Newtonian (e.g. power law) models, needs additional (and earlier) acknowledgement and discussion. The first two examples of the new restoration technique use forward models that also assume Newtonian fluids. These are insightful; however, ideally, I would like to see a restoration of a forward model that uses more realistic rheology for the forward model.*

Additional acknowledgement and discussion of the simplification introduced by (linear) Newtonian viscosity has been added to the introduction. The restoration of a more realistic model (coming from the numerisation of an analogical model) is on course but needs further work and will be published in a following article.

*\* Second, I believe that the third example (Section 4.3) requires additional explanation and discussion. 1. It's not clear how the geometry was constructed. To what extent is this "image" an interpretation of real data vs. based on a model? How was it generated? The general reader should not have to read the reference to understand this model which is critical to this manuscript. 2. Also, why use a stochastically generated diapir rather than a previously published interpretation of a real structure? As a geologist, I would be more comfortable with an example that used a real subsurface structure than a stochastic model.*

The introduction of this model has been changed to be more thorough, as it was indeed not clear that the model actually comes from a seismic image. The model itself, as an interpretation of the seismic image, was generated through a stochastic code to determine the best position of the salt-sediment interface.

*\* Further, the results of this section are very interesting, and probably warrant additional discussion. 1. It makes sense that the system tends toward a state that is in*

*mechanical equilibrium (thus a flat salt-sediment interface). It would be nice to know that the restoration path is valid, too. 2. I'm having trouble understanding what are the geologic implications of the restored images in which synkinematic sediments are not removed. What is or is not representative of past state? What parts of the restored images should I focus on (and what should I not focus on)? There are significant differences between the models in the shallower section, but perhaps the authors do not discuss because they consider it geologically irrelevant. 3. It is important to note that the loading of shallower (younger) sediment is not removed and thus the stress state driving restoration in the past is incorrect. 4. A video (or several key frames) of the preferred restoration as it progresses back in time might add value in addition to showing only the final state of each.*

Additional discussion of the results was added for this model to answer your answer. To summarize some key-points: the sediment and salt layers couldn't be restored to a completely flat state as the stress state inside the model are indeed incorrect for various reasons. A notable point of this result is that this salt diapir example is the result of upbuilding and not downbuilding. A video of the prefered restoration was also added, its link is available at the end of the conclusion, in the "video supplement" part.

* *The authors discuss the ability of this method to discuss faulted structures, and it seems the numerical implementation is ready. It would be nice to include an example.*

After considering the reviews and commentaries on the manuscript, in order to add more value and explain further the possible applications of the restoration idea, another simple example of a model containing two faults and a free surface was added and discussed in the manuscript.

* *The discussion of the numerical implementation (Section 3) is lengthy, and this detracts from the focus on structural restoration. Further, there are many prior implementations of Stokes flow using particle-in-cell methods. I recognize that the numerical implementation was much of the effort, but consider if it would be appropriate to con-*

[Figure]

*dense this section and move the details to the Appendices (along with the validation examples). This could provide space in the manuscript for additional examples and discussion.*

The presented implementation is indeed far from being the first of its kind, but those implementations are not always presented precisely, if at all. In this context, we feel that a clear presentation may help other authors. Additionally, the adaptive grid refinement part of the code is quite specific and has a great impact on the results that could be obtained. In this light, we chose to keep the discussion on the numerical implementation.

* *Finally, following are a few more technically specific comments. Figure 1: Verify that the velocity fields (BC) correspond to this sketch (A). I think that these velocity fields represent a single wavelength perturbation of the material contrast in the horizontal dimension, but the sketch shows two wavelengths perturbation. In other words, for this sketch, there should be four convection cells, not two, and material should be flowing up at the side boundaries in the forward sense.*

There was indeed a mistake in this figure, it has been corrected in the revised version of the manuscript.

* *Paragraph beginning line 279: Use of the term "weld" in the sense of restoration is confusing. The diapir is restored, and sediment is juxtaposed against sediment where there was originally no salt. This is not a weld in the geologic sense. To avoid confusion, I would recommend finding an alternate description of this feature of restoration.*

The term could indeed be misleading, and we chose to replace it with "salt scar".

* *Paragraph beginning line 321: The authors provide two solutions to the rock-air (or -water) interface problem: sticky air or the free surface. They go on to explain the issues with a free surface in some detail, but do not offer further discussion of the sticky-air solution. If it is a viable solution, why not demonstrate it?*

The sticky air method was still being implemented in the code at the publishing of the manuscript. Since then, the implementation showed that the method didn't stabilize the observed instabilities. This point has been added in the revised version of the article.

*\* Reference to Medwedeff., et al.2016 (abstract) is now available in peer reviewed paper(Lovely, Jayr  Medwedeff, AAPG Bulletin, 2018)*

The reference has been updated in the manuscript and bibliography.

*\* Lines 65-67: I don't understand why large deformation and potential remeshing may limit the value for interpretation validation. Would remeshing not be OK, so long as key structural elements (e.g. faults and horizons) are preserved?*

Indeed, this part of the introduction was not clear and has been updated in the manuscript.

*\* Line 111: Should a reference be provided for CFL condition?*

A reference to the original paper presenting the CFL condition was added.

*\* Lines 136-139: Another reason not to solve the thermal equations is that diffusion maybe important at geologic time scales, and it is not reversible.*

Good point, which was added to the manuscript there.

———————————————

---

## Editor Comment (EC1) · Patrice Rey (Editor) · 5 Aug 2020

Hi Melchior,

In your document "AC2: 'Answer to all comments', Melchior Schuh-Senlis, 04 Aug 2020", there is a small error on page C3.

Instead of "# Reviewer 1 (Frantz Maerten)" this should be "# Reviewer 2 (Peter Lovely):"

Thanks

Patrice

---

## Author Comment (AC4) · 5 Aug 2020

Hi Patrice,

Yes, I realized the mistake just after posting the commentary. I tried to modify it, but couldn't find any option allowing the modification of posted commentaries, so I sent an e-mail to editorial@copernicus.org for the correction of this mistake. Is there any other way to proceed for the correction of such mistakes ?

Thanks in advance, and sorry for the inconvenience,

Melchior